# Agent-Omit: Adaptive Context Omission for Efficient LLM Agents

**Yansong Ning** [1]  **Jun Fang** [2]  **Naiqiang Tan** [2]  **Hao Liu** [1]

## Abstract

Managing agent context (e.g., thought and observation) during multi-turn agent-environment interactions is an emerging strategy to improve agent efficiency. However, existing studies treat the entire interaction trajectories equally, overlooking that the thought necessity and observation utility vary across turns. To this end, we first conduct quantitative investigations into how thought and observation affect agent effectiveness and efficiency. Based on our findings, we propose **Agent-Omit**, a unified training framework that empowers LLM agents to adaptively omit redundant thoughts and observations. Specifically, we first synthesize a small amount of cold-start data, including both single-turn and multi-turn omission scenarios, to fine-tune the agent for omission behaviors. Furthermore, we introduce an omit-aware agentic reinforcement learning approach, incorporating a dual sampling mechanism and a tailored omission reward to incentivize the agent's adaptive omission capability. Theoretically, we prove that the deviation of our omission policy is upper-bounded by KL-divergence. Experimental results on five agent benchmarks show that our constructed Agent-Omit-8B could obtain performance comparable to seven frontier LLM agents, and achieve the best effectiveness-efficiency trade-off among seven efficient LLM agent methods. Our code and data are available at https://github.com/usail-hkust/Agent-Omit.

## 1. Introduction

Large language model (LLM) agents solve complex real-world tasks through autonomously interacting with external tool/resources with interleave thought, action, and environment observation (Su et al., 2025). Recently, the paradigm

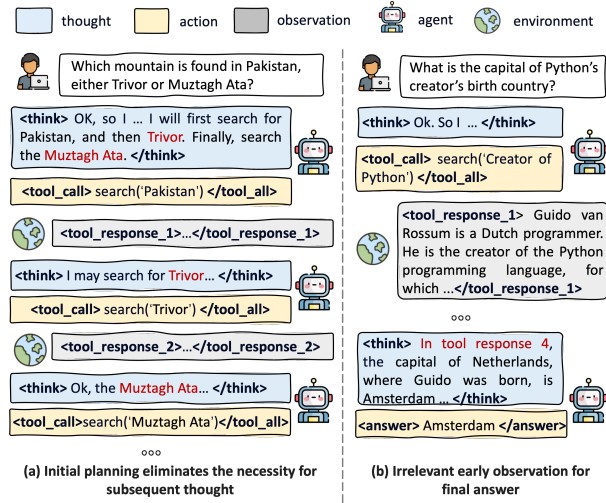

*Figure 1.* Illustrative examples of how thought necessity and observation utility vary across turns. (a) Initial planning (e.g., **search for Trivor and Muztagh Ata**) already determines the subsequent tool call action, making follow-up thought redundant; (b) Observations from early turns are unuseful in the last turn, because **only tool response in turn 4 is used for the answer summarization**.

of agentic reinforcement learning (RL) has further pushed the boundaries of this field (Shang et al., 2025). By interacting with environments and iteratively refining agent policy based on task-specific feedback, agentic LLMs like Kimi-K2 (Team et al., 2025a) and DeepSeek-V3.2 (Liu et al., 2025a) have demonstrated remarkable capabilities in a wide of domain applications, such as deep search, web navigation, digital game, embodied decision-making, and scientific discovery (Xi et al., 2025). Despite these advances, these agents often suffer in generating redundant thought even for simple tool-call actions and accumulating excessive observation context over multiple turns (Wang et al., 2025), limiting their efficiency and practical applicability.

Recent efforts have explored various strategies to mitigate these overheads. Overall, they can be categorized into three paradigms: *Thought Management (TM)*, *Observation Management (OM)*, and *Thought&Observation Management (TOM)*. *TM* and *OM* are the most direct strategies, focusing on compressing thought or pruning historical observations. For example, ToolLight (Chen et al., 2025b) and DEPO (Chen et al., 2025a) employ fine-tuning to compress

---

[1]AI Thrust, The Hong Kong University of Science and Technology (Guangzhou) [2]Didichuxing Co. Ltd. Correspondence to: <liuh@ust.hk>.

*Proceedings of the 43rd International Conference on Machine Learning*, Seoul, South Korea. PMLR 306, 2026. Copyright 2026 by the author(s).

the thought token length, while Observation-Mask (Lindenbauer et al., 2025) and DeepMiner (Tang et al., 2025) use the heuristic strategy to selectively omit historical observations. To address both simultaneously, *TOM*-based approaches like MEM-Agent (Yu et al., 2025) and ReSum (Wu et al., 2025) employ LLM-based summarization tools to jointly compress thought and observation into a concise context. However, these studies **tend to equally compress or modify the entire interaction trajectory**, overlooking that the influence of thoughts and observations can vary across different turns. Our motivation stems from a key assumption: the necessity of thoughts and observations utility varies across turns. As illustrated in Figure 1(a), an agent's initial high-level planning often makes follow-up reasoning thoughts redundant once the execution is clear. Similarly, early-turn observations, while necessary for initial reasoning, usually become irrelevant noise during the final answer summarization phase. These limitations hinders the development of more flexible and efficient agents.

To address this, we first conduct a quantitative investigation into how thought and observation affect agent effectiveness and efficiency. Using Monte Carlo rollouts (Snell et al., 2024), we observe that not all turns contribute equally to task success, and selectively omitting redundant thought/observation can significantly reduce token cost without sacrificing accuracy. Building upon these insights, we propose **Agent-Omit**, a unified framework that empowers LLM agents to adaptively omit redundant thoughts and observations. Our approach consists of two stages: (1) *Agent Omission Behavior Synthesis*, which constructs both of single-turn and multi-turn omission cold-start data to provide initial supervision for efficient agentic reasoning patterns; and (2) *Omit-Aware Agentic Reinforcement Learning*, which incorporates a dual sampling mechanism and a tailored omission reward to progressively improve the agent's capability for adaptive thought/observation omission. Finally, we theoretically prove that the deviation of our omission policy is upper-bounded by the KL-divergence.

We evaluate Agent-Omit on five benchmarks, including DeepSearch, WebShop, TextCraft, BabyAI, and SciWorld. Experimental results show that Agent-Omit-8B achieves accuracy comparable to seven frontier LLMs (e.g., DeepSeek-R1-0528 and o3), while substantially reducing token cost. Moreover, when applied to Qwen3-8B, Agent-Omit consistently outperforms seven efficient LLM agent construction methods, achieving the best effectiveness–efficiency trade-off. Further analysis reveals that the trained agent can adaptively omit 3–4 rounds of thought/observation, where omissions predominantly occurring in intermediate turns, aligning with our empirical findings.

In summary, our key contributions are threefold: (1) We establish a unified thought and observation analysis framework

for LLM agent, quantitatively proving that omission mechanism can improve agent efficiency without sacrificing their effectiveness. (2) We propose Agent-Omit, a framework that integrates omit behavior synthesis with omit-aware agentic RL to train agents capable of adaptive context management. (3) Extensive experiments and theoretical analysis demonstrate the effectiveness of our proposed approach, offering a new paradigm for efficient LLM agents.

**Conflict of Interest Disclosure.** The authors declare no potential financial conflicts of interest related to the work presented in this paper.

## 2. Preliminary

To formalize our investigation, we first define the interaction between an LLM agent and an environment $\mathcal{E}$ over multiple turns. At each turn $t$, the state of the interaction is characterized by three components:

**Definition 2.1. Thought ($\tau_t$).** *A thought is the chain-of-thought reasoning where the agent analyzes current context, plans subsequent actions, or reflects on historical feedback.*

**Definition 2.2. Action ($a_t$).** *An action is an operation chosen by the agent from a predefined action space, such as invoking a tool or generating the final response to the user.*

**Definition 2.3. Observation ($o_t$).** *An observation, denoted as $o_t = \mathcal{E}(a_t)$, is the feedback returned by the environment after executing action $a_t$. It provides the necessary grounding for the next interaction turn.*

## 3. Analysis of Thought and Observation on Agent Effectiveness and Efficiency

We hypothesize that the necessity of thoughts and observations utility is not equal but rather turn-dependent:

**Assumption 3.1. Thought necessity varies across turns.** *Not all turns require detailed reasoning processes. While complex steps necessitate in-depth thought for planning or reflection, serval intermediate turns can often be executed directly without extra thought, e.g., execute a simple tool-call action $a_t$ derived from a prior plan $\tau_{<t}$.*

**Assumption 3.2. Observation utility varies across turns.** *Not all turns require entire observation context. As the interaction trajectory grows longer, certain past observations $o_{<t}$ will become redundant or irrelevant to the current action $a_t$, e.g., outdated search results cannot contribute to the final answer summarization.*

Then, we conduct quantitative analysis to validate the aforementioned assumption.

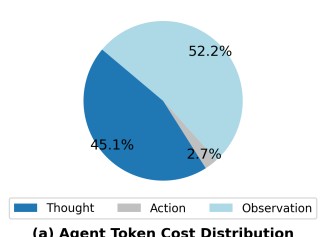
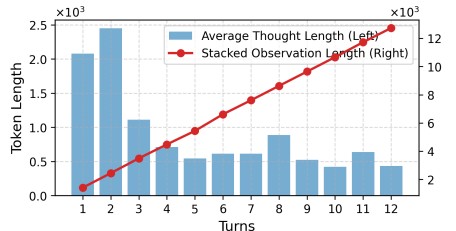
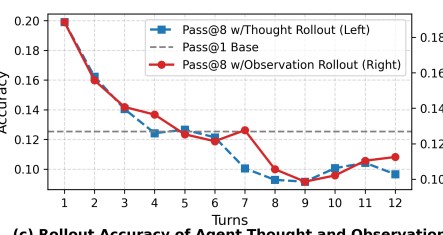

(a) Agent Token Cost Distribution     (b) Token Length of Agent Thought and Observation     (c) Rollout Accuracy of Agent Thought and Observation

*Figure 2.* Quantitative analysis of how thought and observation affect agent efficiency and effectiveness across interaction turns on WebShop environment using Qwen3-8B.

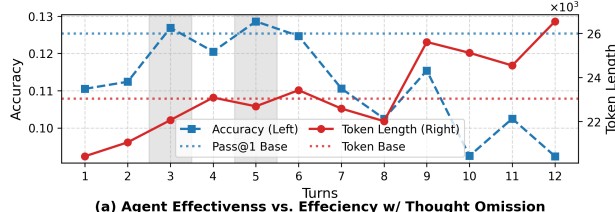
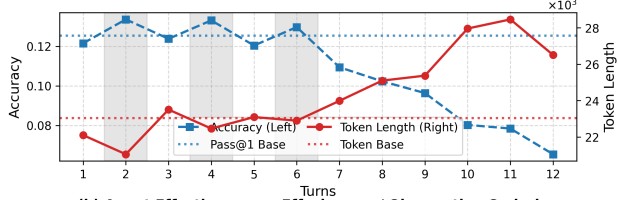

(a) Agent Effectivenss vs. Effeciency w/ Thought Omission     (b) Agent Effectivenss vs. Effeciency w/ Observation Omission

*Figure 3.* The effect of thought and observation omission on agent effectiveness and efficiency across interaction turns on the WebShop environment using the Qwen3-8B. For each turn $t \in \{1, \ldots, 12\}$, we omit the thought $\tau_t$ (or observation $o_t$) at that specific turn and let the agent continue the remaining reasoning trajectory to task completion. Results are averaged over all 200 randomly sampled test cases. The grey shaded regions indicate turns where omission reduces token cost without sacrificing task accuracy (i.e., the accuracy $\geq$ Pass@1 Base and the token length $<$ Token Base).

### 3.1. Quantitative Analysis

Using Qwen3-8B on the WebShop (Yao et al., 2022) environment, we examine the turn-wise token costs caused by thought and observation, and their respective contributions to task accuracy. The key findings, illustrated in Figure 2, are summarized as follows:

- **Agent Bottlenecks on Thought & Observation.** As shown in Figure 2(a), the token cost of agent is dominated by thought (45.1%) and observation (52.2%). Actions account for only 2.7%, indicating that the major bottleneck for agent efficiency lies in the thought and environmental observations, rather than the execution of actions itself.
- **Impact on Efficiency.** Figure 2(b) further shows how these token costs evolve over interaction turns. We find that the *thoughts are front-loaded*, with heavy token consumption in early turns (e.g., Turns 1-2) for high-level planning. In contrast, the *observation exhibits a linear growth* due to the stacking mechanism, leading a heavy context burden for later turns. Overall, these phenomenon demonstrate that **the impact of thought/observation on agent efficiency varies across turns**.
- **Impact on Effectiveness.** Third, we perform Monte Carlo rollouts at each turn to measure the contribution of thought/observation to task success. As shown in Figure 2(c), while early thoughts and observations are critical for high accuracy (Pass@8), the accuracy gain diminishes rapidly as the interaction progresses, with later turns often falling below the Pass@1 baseline. These observation

demonstrates that **not all thoughts/observations at each turn contribute equally for agent effectiveness**.

**Key Insight.** Based on these observations, a natural question arises: *Why not selectively omit the thoughts and observations that consume excessive token length but fail to contribute to accuracy?* To answer this question, we propose a thought and observation omission mechanism to investigate whether such selective omission can be achieved without degrading performance.

### 3.2. Thought and Observation Omission Mechanism

To test whether these redundant thoughts/observations can be safely omitted, we conducted a controlled intervention: given an interaction trajectory until turn $t$, we explicitly omit either the generated thought $\tau_t$ or the observation $o_t$, and prompt the agent to continue the reasoning trajectory to derive the final answer. We then compare the resulting task accuracy and total token cost with that of a base agent that continues the trajectory without omission. The overall results are shown in Figure 3. We summarize the key findings observed as follows:

- **Thought Omission.** Figure 3(a) shows that omitting intermediate reasoning steps yields improved accuracy and reduced token usage. However, omitting thoughts during the initial or final turns proves detrimental. We attribute this phenomenon to two factors: (1) in the initial turn, explicit reasoning is crucial for high-level planning; (2)

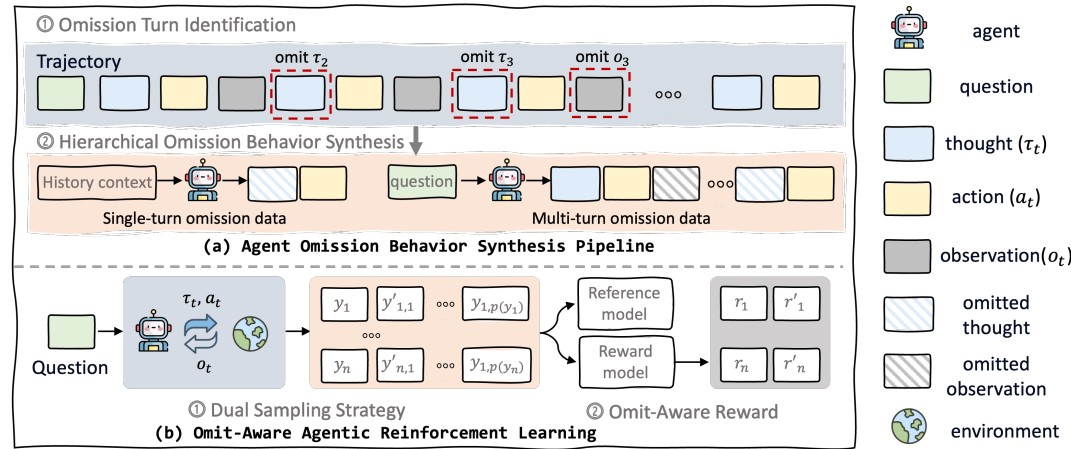

*Figure 4.* Overview of our proposed framework **Agent-Omit**.

in the final turns, removing critical thought disrupts contextual continuity, forcing the agent to generate additional thoughts to recover the lost context.

- **Observation Omission.** Similarly, Figure 3(b) indicates that observations are most omissible in the intermediate turns (Turns 2, 4, 6), where agent's token cost is decreased without sacrificing accuracy. In later turns, observations become indispensable because omitting them causes a significant drop in accuracy. We attribute this to the fact that later observations may contain critical information. When this is missing, the agent not only fails to solve the task but also tends to generate excessive reasoning tokens to bridge the information gap.

**Motivation for Adaptive Omission.** The existence of grey regions in Figure 3 demonstrates that substantial efficiency gains are possible without sacrificing performance. These findings reveal a crucial insight: *It is possible to reduce token costs without sacrificing accuracy, but only if the omission is applied selectively.* However, because the optimal omission points are dynamic and task-dependent, a static heuristic is insufficient. This motivates the need for **Agent-Omit**, a framework that learns an adaptive policy to identify and omit redundant context during interaction.

## 4. Efficient LLM Agents with Agent-Omit

Building on our analysis, we define the agent's objective as balancing task success with minimal context overhead. We formalize this as a selective decision-making process:

**Problem 1.** *Adaptive Thought and Observation Omission. Given a question $q$ and the agent interaction history $\{\tau_1, a_1, o_1, \tau_2, a_2, o_2, \ldots, \tau_{t-1}, a_{t-1}, o_{t-1}\}$ at turn $t$, the agent generates a thought $\tau_t$ and action $a_t$:*

$$\{\tau_t, a_t\} = \pi_\theta \left( q, \{\tau_1, a_1, o_1, \ldots, \tau_{t-1}, a_{t-1}, o_{t-1}\} \right), \quad (1)$$

*where $\pi_\theta$ denotes the policy of the agent, and $\tau_t$ is the*

*thought process, which the policy may adaptively reduce to an empty string $\emptyset$ if reasoning is redundant. The action consists of the environment action and an observation omission set $\Gamma_t \subseteq \{1, \ldots, t-1\}$, identifying specific historical turns whose observations $o_i$ are irrelevant and should be omitted.*

To this end, we propose **Agent-Omit**. As illustrated in Figure 4, Agent-Omit achieve this through a two-stage optimization: (a) synthesizing cold-start data to fine-tune agent to establish the omission format, and (b) an omit-aware agentic RL method for agent training to achieve adaptive thought/observation omission.

### 4.1. Agent Omission Behavior Synthesis for Cold Start

To bridge the gap between generalist LLMs and omission-aware agents, we construct a synthetic cold-start dataset that explicitly teaches the agent (i) how to execute omission, and (ii) how to continue the reasoning process under omitted historical context. Specifically, it consists of three phases: omission turn identification, hierarchical omission behavior synthesis, and cold-start training.

**Omission Turn Identification.** Instead of relying on heuristics (Tang et al., 2025), we identify "omittable" turn by performing rollout over trajectories. Specifically, we traverse the interaction trajectory in a forward manner and evaluate each turn $t$ by explicitly omitting either the generated thought $\tau_t$ or the observation $o_t$. After the omission, the agent is prompted to continue the remaining reasoning process until the final answer is produced. If the omission leads to reduced token without degrading task accuracy, we mark the corresponding turn as omittable. For instance, in the example shown in Figure 4(a), we identify $\tau_2, \tau_3$ and $o_3$ as redundant turn that can be omitted.

**Hierarchical Omission Behavior Synthesis.** Based on the identified turns, we build the synthetic dataset. We adopt a

hierarchical approach: first constructing single-turn omission samples to teach the agent omission format, and then expanding into multi-turn scenarios where agent should continue the reasoning process under omitted historical context:

- **Single-Turn Omission.** As shown in Figure 4(a), the goal here is to teach agent how to execute omission. Similar with Qwen3 thinking mode fusion paradigm (Yang et al., 2025), we manually devise a omission system prompt to guide agent. Guided by the tailored system prompt, the agent learns two distinct omission behaviors: i) thought omission, where the agent generates an empty thought $\emptyset$ within specific tokens (i.e, `<think> </think>`); and ii) observation omission, where the agent outputs a strategy command (i.e., `<omit_tool_response_N_...>` `</omit_tool_response_N_...>`) to explicitly trigger the removal of historical observation set $\Gamma \subseteq \{1, \ldots, t-1\}$ from the context.

- **Multi-Turn Omission.** We then construct multi-turn full trajectories by replacing the original thoughts and observations with their corresponding omission behaviors. This setup forces the agent to maintain reasoning continuity even when the interaction history is "omitted". By training on these trajectories, the agent learns to avoid context-lost (Laban et al., 2025), where it unnecessarily attempts to recover information that has been properly omitted.

**Cold-Start Training.** Finally, we perform full-parameter fine-tuning using the synthesized dataset $\mathcal{D}_{single}$ and $\mathcal{D}_{multi}$ with standard language modeling loss:

$$\mathcal{L} = -\mathbb{E}_{(x,y) \sim D_{single} \cup D_{multi}} \left[ \log \mathcal{P}_{\pi_\theta}(y \mid x) \right]. \quad (2)$$

To ensure the policy learns only the agent-generated token, we apply a loss mask (Jin et al., 2025) mechanism to all environmental observations $o_i$ during training.

### 4.2. Omit-Aware Agentic Reinforcement Learning

While the SFT initializes omission format, it relies on synthetic data and cannot generalize to dynamic interactions. To address this, we propose a omit-aware agentic RL framework to incentivize omission capability. It consists of a dual sampling strategy, an omission-aware reward mechanism, and multi-objective policy learning.

**Dual Sampling Strategy.** A critical challenge in training omission policy is the "context change" problem: once an observation is omitted, the agent no longer "sees" the information it used to make that decision. Since current agentic RL training (Zhang et al., 2025) relies on a single post-hoc trajectory, the agent never observes the pre-omission context, making omission policy unlearnable. To address this, we decouple sampling into two process:

- **Full Trajectory ($y$).** Following existing RL paradigm (Feng et al., 2025), we sample a complete multi-turn in-

teraction trajectory where the agent's omission action are executed. This stream evaluates the overall efficiency and final task success of the interaction.

- **Partial Trajectory ($y'$).** For each turn where an omission is triggered, we treat the current context and the agent's single-turn thought/action as a partial trajectory. For each full trajectory $y$, this process yields a variable number, denoted as $p(y)$, of partial trajectories. This design ensures that agent can learn the omission policy conditioned on the pre-omission context, instead of learning from already-compressed trajectories.

**Omit-Aware Reward.** We design two types of reward function to balances task correctness and token efficiency:

- **Task Reward ($R_{task}$).** For both full and partial trajectories, the primary objective is accuracy. We assign a task reward based on the correctness of the full trajectory or the derived final answer from the partial trajectory.

- **Omission Reward ($R_{omit}$).** To explicitly encourage token reduction, we propose the omision reward for full trajectory. It is calculated as the ratio of saved tokens:

$$R_{omit} = Tok(\tau_{omitted})/Tok(y) + Tok(o_{omitted})/Tok(y), \quad (3)$$

where $Tok(.)$ is the token count function. Critically, $R_{omit}$ is set to 0 if $R_{task} = 0$, ensuring the agent cannot achieve high efficiency through reward hacking.

**Multi-Objective Policy Learning.** We optimize the agent policy $\pi_\theta$ to maximize the expected reward across both partial and full trajectory. We employ Group Relative Policy Optimization (GRPO) (Shao et al., 2024) to stabilize training process. Overall, the unified training objective is formulated as follows:

$$\max_{\pi_\theta} \mathbb{E}_{x \sim \mathcal{D}, \left\{ y_i, \{y'_{i,j}\}_{j=1}^{p(y_i)} \right\}_{i=1}^n \sim \pi_\theta(\cdot|x)} \left[ \frac{1}{n} \sum_{i=1}^n \left( r(x, y_i) \right. \right.$$

$$\left. \left. + \frac{1}{p(y_i)} \sum_{j=1}^{p(y_i)} r'(x, y'_{i,j}) \right) \right] - \beta \mathbb{D}_{KL} \left[ \pi_\theta \| \pi_{ref} \right], \quad (4)$$

where $n$ is the rollout size, $p(y_i)$ is the number of partial trajectory corresponding to each full trajectory, $\beta$ is the coefficient for the KL-divergence penalty, $r(.) = (1 - \mu) * R_{task} + \mu * R_{omit}$ is the reward reweighted function for the full trajectory $y$, and $r'(x, y'_{i,j}) = R_{task}$ denotes the task reward for partial trajectory $y'_{i,j}$, computed by continuing the reasoning from the partial context to derive a final answer. Based on best empirical practice, we set $\mu = 0.2$. We follow Verl (Sheng et al., 2024) to utilize a loss mask for environment observation (Sheng et al., 2024).

# 5. Theoretical Analysis

In this section, we theoretically characterize the quality of the learned omission policy. Specifically, we ask: *how much does the agent's effectiveness and efficiency deviate from the optimal omission strategy?* We show that this deviation is upper-bounded by the distributional distance between the learned policy $\pi_\theta$ and the optimal omission policy $\pi^*$, measured via KL-divergence in Equation 4. This provides a principled guarantee that as $\pi_\theta$ better approximates $\pi^*$ during training, the omission quality monotonically improves.

We first establish the deviation in effectiveness (task reward) and efficiency (token cost) between our generated omission trajectory $y$ and the optimal one $y^*$:

**Assumption 5.1** (Semantic Lipschitz Continuity). Assume that the agent's task accuracy $R(y)$ and token cost $C(y)$ are Lipschitz continuous with respect to the semantic distance in the trajectory embedding space:

$$|R(y^*) - R(y)| \leq K_r \cdot \mathrm{d}(y^*, y), \tag{5}$$

$$|C(y^*) - C(y)| \leq K_c \cdot \mathrm{d}(y^*, y), \tag{6}$$

where $\mathrm{d}(\cdot, \cdot)$ denotes the distance metric in the semantic space, $K_r$ and $K_c$ are Lipschitz constants (Hager, 1979).

While Assumption 5.1 ensures that similar trajectories yield similar outcomes, Agent-Omit essentially learns a policy $\pi_\theta$ that approximates the optimal omission policy $\pi^*$. The total omission error can be modeled as the policy distributional shift during RL training. Therefore, we further derive the bound of the omission error:

**Theorem 5.2** (Bounded Omission Error). *The expected deviation in effectiveness and efficiency between the learned policy $\pi_\theta$ and the expected policy $\pi^*$ is upper-bounded by the KL-divergence $\mathrm{KL}(\pi^*, \pi_\theta)$:*

$$|\mathbb{E}[R(y^*)] - \mathbb{E}[R(y)]| \leq \delta_r + K_r' \cdot \mathrm{KL}(\pi^*, \pi_\theta), \tag{7}$$

$$|\mathbb{E}[C(y^*)] - \mathbb{E}[C(y)]| \leq \delta_c + K_c' \cdot \mathrm{KL}(\pi^*, \pi_\theta), \tag{8}$$

*where $\delta_r, \delta_c$ are irreducible approximation errors, $K_r'$, $K_c'$ are the scaled Lipschitz constant.*

Theorem 5.2 suggests that, as the reinforcement learning objective in the Eq. 4, the KL divergence between the learned policy $\pi_\theta$ and the optimal omission policy $\pi^*$ could be gradually minimized. Therefore, the agent's accuracy and token cost converge to the optimal omission frontier, limited only by the approximation errors. Detailed proofs are provided in Appendix A.

# 6. Experiments

## 6.1. Experimental Setup

**Agent Environments.** As shown in Table 1, to evaluate Agent-Omit across diverse reasoning and interaction pat-

*Table 1.* Statistics of agent environments and training data.

| Dataset | Training Set | | Test Samples | Maximum Turns / Tokens |
|---|---|---|---|---|
| | Cold Start | RL | | |
| DeepSearch | 3,257 | 2,000 | 400 | 8 / 32K |
| WebShop | 2,134 | 2,000 | 200 | 12 / 32K |
| TextCraft | 2,911 | 374 | 100 | 20 / 32K |
| BabyAI | 2,758 | 810 | 90 | 10 / 32K |
| SciWorld | 2,514 | 2,120 | 200 | 10 / 32K |

terns, we conduct experiments on five distinct domains following the AgentGym-RL (Xi et al., 2025):

- **Information Search.** We use **DeepSearch** (Jin et al., 2025), requiring agents to invoke search engines to resovle knowledge-intensive queries.
- **Web Navigation.** We employ **WebShop** (Yao et al., 2022), where agents navigate e-commerce website to perform structured attribute extraction and purchasing.
- **Digital Games.** We evaluate on **TextCraft** (Prasad et al., 2024), a Minecraft-inspired environment for testing long-horizon planning and recipe crafting.
- **Embodied Control.** We include **BabyAI** (Chevalier et al., 2018), which focuses on instruction following and navigation within partially observable grid-worlds.
- **Scientific Discovery.** We use **SciWorld** (Wang et al., 2022) to assess complex reasoning in physical simulations, involving hypothesis testing and experiment design.

**Baselines.** We benchmark our approach against two categories of baselines to verify both absolute Pass@1 accuracy and token efficiency gains:

- **Frontier LLM Agents.** We compare against state-of-the-art models including DeepSeek-R1-0528 (Guo et al., 2025a), DeepSeek-V3.2 (Liu et al., 2025a), OpenAI o3/o4-mini (OpenAI, 2025), Qwen3-235B-A22B, Qwen3-Next-80B-A3B and Qwen3-32B (Yang et al., 2025) as general-purpose backbones.
- **Efficient Agents Construction Methods.** To validate the efficiency benefits of Agent-Omit, we compare against three paradigms of efficiency optimization:
  - **Thought Management**: We include Thinking-Retention (Liu et al., 2025a), which directly prunes historical thought; DEPO (Chen et al., 2025a) and Tool-Light (Chen et al., 2025b), which compresses thought tokens via post-training.
  - **Observation Management**: We compare against Observation-Mask (Lindenbauer et al., 2025) strategy and the heuristic sliding-window approach proposed by DeepMiner (Tang et al., 2025).
  - **Thought&Observation Management**: We evaluate summarization-based methods, specifically MEM-Agent (Yu et al., 2025) and ReSum (Wu et al., 2025), which actively summarize the interaction context.

*Table 2.* Comparison between Agent-Omit-4B/8B and existing frontier LLM agents. The grey-shaded rows represent non-reasoning mode, for which lower token consumption is reasonable. Token statistics for closed-source models (e.g., OpenAI o3/o4-mini) are unavailable due to serious API restrictions.

| Models | DeepSearch | | WebShop | | TextCraft | | BabyAI | | SciWorld | |
|---|---|---|---|---|---|---|---|---|---|---|
| | Pass@1 ↑ | Avg Tok. ↓ | Pass@1 ↑ | Avg Tok. ↓ | Pass@1 ↑ | Avg Tok. ↓ | Pass@1 ↑ | Avg Tok. ↓ | Pass@1 ↑ | Avg Tok. ↓ |
| DeepSeek-R1-0528 | 25.25 | 6,412 | 19.37 | 11,308 | 83.00 | 7,435 | 81.74 | 9,578 | 13.23 | 18,288 |
| DeepSeek-V3.2 | 27.25 | 2,175 | 9.40 | 3,104 | 77.00 | 1,768 | 80.20 | 1,166 | 10.09 | 1,642 |
| OpenAI o3 | **31.50** | - | 12.43 | - | 73.00 | - | 78.54 | - | 10.08 | - |
| OpenAI o4-mini | 25.60 | - | 14.69 | - | 75.00 | - | 75.44 | - | 8.59 | - |
| Qwen3-235B-A22B | 22.30 | 7,225 | 13.25 | 7,633 | 72.00 | 8,246 | 74.46 | **6,542** | 9.25 | 9,166 |
| Qwen3-Next-80B-A3B | 9.25 | 3,127 | 8.96 | 2,361 | 69.00 | 2,697 | 67.87 | 1,953 | 3.37 | 4,093 |
| Qwen3-32B | 19.00 | 6,640 | 11.31 | 11,872 | 59.00 | 16,294 | 64.10 | 19,489 | 3.57 | 19,211 |
| Agent-Omit-4B-SFT | 9.23 | 6,346 | 10.25 | 9,543 | 62.00 | 13,256 | 61.27 | 14,281 | 5.25 | 16,367 |
| Agent-Omit-4B-RL | 21.70 | **3,312** | 16.31 | **7,378** | 74.00 | **5,341** | 72.42 | 7,728 | 10.14 | **8,805** |
| Agent-Omit-8B-SFT | 22.25 | 6,153 | 14.43 | 11,376 | 72.00 | 11,125 | 72.25 | 12,265 | 8.54 | 12,256 |
| Agent-Omit-8B-RL | 26.56 | **4,356** | **23.57** | 8,764 | **87.00** | **7,328** | **84.36** | 6,643 | **18.45** | 9,643 |

**Implementation Details.** We validate the proposed Agent-Omit framework—comprising a sequential Cold Start phase and an agentic RL module—using Qwen3-4B and Qwen3-8B backbones. This process yields four distinct models: Agent-Omit-4B/8B-SFT and Agent-Omit-4B/8B-RL. For the SFT cold-start, we systhesize about 2-4K training samples, with a learning rate of 5e-6, over about 3 epochs. For the RL training, we follow the AgentGym-RL settings and employ our proposed omit-aware policy optimization. During RL, we use a learning rate of 5e-7. The maximum response length is set to 32k tokens, and the maximum interaction turns varies according to task, as shown in Table 1. Training the Qwen3-8B and Qwen3-4B models requires 8 and 4 NVIDIA A100 GPUs, respectively.

More detailed agent environment configurations, system prompts, data synthesis, evaluation protocols, and training hyperparameters are provided in Appendix B.

### 6.2. Main Results

**Comparison with Frontier LLM Agents.** As presented in Table 2, our Agent-Omit-8B-RL demonstrates exceptional capability, achieving the highest Pass@1 score across WebShop, TextCraft, BabyAI, and SciWorld, while remaining highly competitive on DeepSearch with scores only marginally below the current state-of-the-art. Regarding to the efficiency, Agent-Omit-8B-RL significantly outperforms reasoning models such as DeepSeek-R1-0528 and Qwen3-32B by requiring substantially fewer tokens to achieve better accuracy. While it incurs a slightly higher token cost compared to non-reasoning mode like DeepSeek-V3.2 and Qwen3-Next-80B-A3B, this trade-off yields substantial gains in task accuracy. Notably, we find that the RL stage is pivotal, it not only further enhance model accuracy during initial SFT, but also actively improve the agent's efficiency, leading to superior cost-effectiveness.

**Comparison with Efficient Agents Methods.** As shown

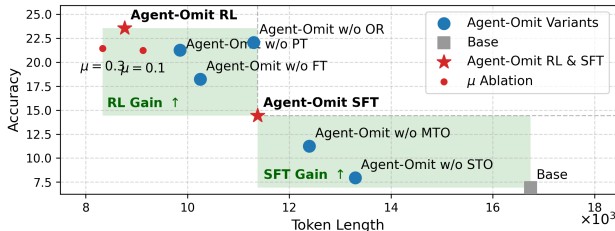

*Figure 5.* Pass@1 accuracy of Agent-Omit variants on WebShop environment using Qwen3-8B backbone.

in Table 3, on the Qwen3-8B, our method consistently outperforms existing paradigms, achieving the highest Pass@1 task accuracy while incurring the lowest average token cost across all five agent benchmarks. On the one hand, we observe that heuristic-based thought/observation management methods (e.g., Thinking-Retention and Observation-Mask baselines) reduce token usage but suffer from significant accuracy degradation due to the lack of model adaptation. On the other hand, while summarization-based methods such as ReSum achieve a good effectiveness–efficiency trade-off, their accuracy improvements remain limited due to the gap between the agent's internal reasoning and the LLM-based summarizer. Overall, we find that Agent-Omit achieves the lowest token cost while maintaining the highest accuracy, demonstrating great potential.

### 6.3. Ablation Study

To evaluate the contribution of each component in **Agent-Omit**, we conduct an ablation study by removing specific modules and changing hyperparameters during the SFT and RL stages. The variants are denoted as: Agent-Omit w/o STO/MTO (removing Single/Multi-Turn Omission data in SFT stage), Agent-Omit w/o PT/FT (removing Partial/Full Trajectory sampling in RL stage), and Agent-Omit w/o OR (removing Omission Reward from RL training).

*Table 3.* Comparison between Agent-Omit-8B and existing efficient LLM agents construction methods.

| Methods | DeepSearch | | WebShop | | TextCraft | | BabyAI | | SciWorld | |
|---|---|---|---|---|---|---|---|---|---|---|
| | Pass@1 ↑ | Avg Tok. ↓ | Pass@1 ↑ | Avg Tok. ↓ | Pass@1 ↑ | Avg Tok. ↓ | Pass@1 ↑ | Avg Tok. ↓ | Pass@1 ↑ | Avg Tok. ↓ |
| *Qwen3-8B* | | | | | | | | | | |
| Base | 17.75 | 8,281 | 6.93 | 16,741 | 55.00 | 19,587 | 60.81 | 19,162 | 2.47 | 17,052 |
| *+ Thought Management* | | | | | | | | | | |
| Thinking-Retention | 10.25 | 5,234 | 5.26 | 11,264 | 57.00 | 12,274 | 52.56 | 13,368 | 2.75 | 11,206 |
| DEPO | 19.26 | 6,625 | 10.26 | 10,286 | 62.00 | 13,256 | 68.25 | 12,291 | 12.39 | 13,092 |
| Tool-Light | 21.29 | 6,239 | 14.57 | 10,892 | 68.00 | 11,247 | 60.25 | 14,482 | 17.29 | 12,865 |
| *+ Observation Management* | | | | | | | | | | |
| Observation-Mask | 12.76 | 8,821 | 7.29 | 9,954 | 52.00 | 12,892 | 57.74 | 15,255 | 1.39 | 14,398 |
| DeepMiner | 16.29 | 6,032 | 6.82 | 11,367 | 58.00 | 13,263 | 61.23 | 13,329 | 5.28 | 15,395 |
| *+ Thought&Observation Management* | | | | | | | | | | |
| MEM-Agent | 17.62 | 5,381 | 6.62 | 10,011 | 56.00 | 11,299 | 63.08 | 11,296 | 6.62 | 11,577 |
| ReSum | 22.28 | 5,724 | 17.80 | 9,251 | 72.00 | 9,258 | 77.09 | 8,826 | 14.48 | 11,782 |
| *+ Ours* | | | | | | | | | | |
| Agent-Omit-8B-RL | **24.56** | **4,356** | **23.57** | **8,764** | **87.00** | **7,328** | **84.36** | **6,643** | **18.45** | **9,643** |

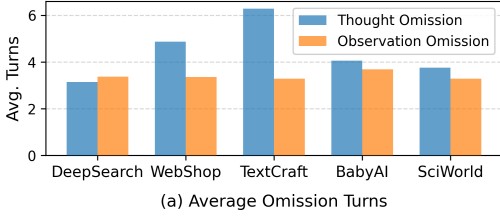

(a) Average Omission Turns

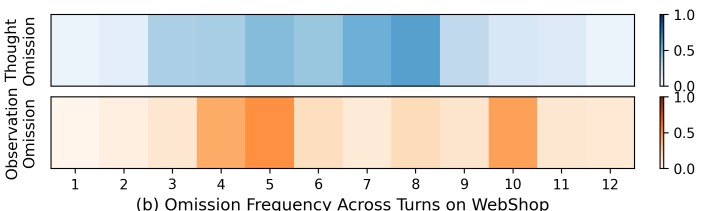

(b) Omission Frequency Across Turns on WebShop

*Figure 6.* Statistics of average omission turns of Agent-Omit-8B-RL and its omission frequency across different turns.

The results are shown in Figure 5, we derive the following key observations:

- **Dual Gain:** Both the SFT and RL stages contribute significantly to the improvements in agent effectiveness and efficiency, as evidenced by the green region in Figure 5.
- **For the SFT phase:** Single-turn omission data proves to be the dominant factor, facilitating the agent's fundamental capability to handle omission behaviors effectively.
- **For the RL stage:** Partial trajectory sampling plays a more critical role for training gains than full trajectory sampling. Furthermore, the omission reward is the primary factor for agent efficiency. Without it, the model fails to achieve more token reduction than SFT version. We also find that changing the reward reweighting factor $\mu$ from 0.2 (either decreasing it to 0.1 or increasing it to 0.3) leads to a worse effectiveness–efficiency trade-off.

Overall, these results verify the necessity and effectiveness of each integrated module within the our framework.

### 6.4. In-Depth Analysis

To understand how the Agent-Omit framework works during the inference stage after training, we further analyze the distribution of omission behavior:

- **Omission Volume:** As shown in Figure 6(a), across tasks, the agent adaptively omits an average of 3 to 4 turns of

redundant thoughts or observations per trajectory.

- **Omission Volume:** Figure 6(b) reveals that omission frequency is not uniform: it peaks during the intermediate turns (turns 3–10). This result indicates that redundant thoughts and observations predominantly occur in the middle phase of the agent-environment interaction process, aligning with our findings in Section 3.

We also provided more detailed agentic RL training analysis and case study in Appendix C.

## 7. Related Work

### 7.1. Efficient LLM Agents

LLM agents have recently transformed diverse application domains (Ning et al., 2025a; Liu et al., 2025b; Ning & Liu, 2024) from static workflows to autonomous agentic planning capable of addressing complex real-world problems (Liu et al., 2026; Team et al., 2025b; Liu et al., 2025c). However, the necessity for multi-turn interactions with external environments often leads to long-context (e.g., redundant thought and stacked environment observations), limiting agent efficiency (Cai et al., 2025; Ning et al., 2025b). Many recent works have made efforts to address this, which can be categorized into three approaches: **(1) Thought Management:** Methods such as WebLeaper (Tao et al., 2025), DEPO, and ToolLight explicitly compress thought processes

(Chen et al., 2025a;a) via fine-tuning and length-aware reward penalties. Additionally, DeepSeek-V3.2 proposes a thinking retention mechanism designed to directly omit historical reasoning content (Liu et al., 2025a). **(2) Observation Management:** Recent works attempt to directly mask prior-turn observations (Lindenbauer et al., 2025) to decrease context length. To enhance flexibility, DeepMiner (Tang et al., 2025) employs a sliding window strategy that selectively omits observations. **(3) Thought & Observation Management:** These approaches rely on summarizing the interaction trajectory (Xiao et al., 2025). For instance, MEM1 (Zhou et al., 2025) and MEM-Agent (Yu et al., 2025) actively summarize interaction history into a concise context, whereas ReSum (Wu et al., 2025) and CAT (Liu et al., 2025d) construct a summarization tool, allowing the agent to actively invoke it for context compression. However, these works alter the entire interaction trajectory, but fail to consider the diverse impact of thought and observation across different interaction turn, limiting the construction of more efficient and flexible agents.

### 7.2. Agentic Reinforcement Learning

Agentic Reinforcement Learning (RL) enables agents to interact with external environments (Ning et al., 2025c) and optimize their policies based on received reward feedback (Shang et al., 2025). Recently, numerous advanced agentic RL algorithms have been proposed to enhance agent capabilities. For instance, ReTool (Feng et al., 2025), Kimi-Researcher (MoonshotAI, 2025), and WebSailor (Li et al., 2025) establish end-to-end RL training frameworks that allow agents to interact within constructed sandboxes (Fang et al., 2025). More recently, works such as AgentEvolver (Zhai et al., 2025) and GenEnv (Guo et al., 2025b) have begun to investigate policy learning for improving sample efficiency or facilitating environment evolution. However, there are still few works focusing on omission policy learning, and lack of tailored agentic RL algorithms.

## 8. Conclusion, Limitation and Future Work

In this work, we establish a turn-level analysis framework that quantitatively evaluates how thoughts and observations influence agent efficiency and effectiveness across different interaction turns. Building on this analysis, we propose Agent-Omit, an experimentally and theoretically grounded framework that enables efficient LLM agent training via omission-data-driven cold-start fine-tuning and omit-aware agentic reinforcement learning. Extensive experiments on five diverse benchmarks show that Agent-Omit allows small-sized LLMs to significantly outperform seven frontier LLM agents and achieve a better effectiveness–efficiency trade-off than seven efficient-agent methods. However, our approach incurs additional training samples during the RL

stage, limiting its scalability. Future work will focus on scaling the omission data synthesis pipeline to the large-scale pre-training stage and extending this training paradigm to larger-size LLMs.

## Acknowledgements

This work was supported by the National Natural Science Foundation of China (Grant No. 62572417, No.92370204), National Key R&D Program of China (Grant No.2023YFF0725004).

## Impact Statement

This work aims to improve the efficiency of LLM-based agents by reducing redundant token cost during multi-turn interactions. By enabling agents to adaptively omit unnecessary thoughts and observations, our approach reduces token consumption and computational cost, which may contribute to lowering the energy footprint of deploying large language model agents. We do not foresee specific negative societal consequences of this work beyond those generally associated with large language models.

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

# A. Theoretical Analysis

In this section, we provide the detailed mathematical derivations for the theoretical analysis of Agent-Omit. We first introduce the standard definition of Lipschitz Continuity, then derive the Semantic Lipschitz Continuity Lemma, and finally prove the Bounded Omission Error Theorem.

## A.1. Preliminaries

**Definition A.1** (Lipschitz Continuity (Hager, 1979))**.** Given two metric spaces $(X, d_X)$ and $(Y, d_Y)$, a function $f : X \to Y$ is called Lipschitz continuous if there exists a real constant $K \geq 0$ such that, for all $x_1, x_2 \in X$:

$$d_Y(f(x_1), f(x_2)) \leq K \cdot d_X(x_1, x_2). \tag{9}$$

Intuitively, this definition guarantees that the function does not oscillate infinitely fast; a bounded change in the input results in a bounded change in the output.

## A.2. Assumption 1: Semantic Lipschitz Continuity

Let function $\phi(\cdot)$ denotes the map function, which transfer the agent's interaction trajectory $y$ into corresponding token embedding. Let $\mathrm{d}(y^*, y) = \|\phi(y^*) - \phi(y)\|$ denote the semantic distance between two trajectories.

**Statement.** The task reward function $R(.)$ and the token cost function $C(.)$ are Lipschitz continuous with respect to the semantic distance $\mathrm{d}(y^*, y)$. Specifically, there exist constants $K_r$ and $K_c$ such that:

$$|R(y^*) - R(y)| \leq K_r \cdot \mathrm{d}(y^*, y), \tag{10}$$
$$|C(y^*) - C(y)| \leq K_c \cdot \mathrm{d}(y^*, y). \tag{11}$$

*Proof.* Let $\mathcal{Y}$ be the space of all possible interaction trajectories. The effectiveness of an agent is measured by the reward function $R : \mathcal{Y} \to [0, 1]$, and efficiency is measured by the token cost function $C : \mathcal{Y} \to \mathbb{R}^+$. The assumption of Semantic Lipschitz Continuity implies that the reward and cost landscapes are smooth with respect to the semantic representation of the trajectory. This is a standard assumption in representation learning, positing that if two trajectories $y$ and $y^*$ have very similar semantic embeddings (i.e., $\mathrm{d}(y^*, y) \to 0$), their resulting task performance and token consumption should also be similar. Therefore, for any two trajectories $y^*, y \in \mathcal{Y}$, the error in their effectivenss and efficency is bounded by the distance in their semantic representations scaled by the Lipschitz constants $K_r$ and $K_c$. $\square$

## A.3. Theorem 1: Bounded Omission Error

Under adaptive omission strategy, we derive the upper bound for the deviation in agent effectiveness and efficiency.

**Statement.** The expected deviation in effectiveness and efficiency between the learned policy $\pi_\theta$ and the expected policy $\pi^*$ is upper-bounded by the their KL-divergence $\mathrm{KL}(\pi^*, \pi_\theta)$:

$$|\mathbb{E}y^* \sim \pi^*[R(y^*)] - \mathbb{E}y \sim \pi_\theta[R(y)]| \leq \delta_r + K_r \cdot \mathrm{KL}(\pi^*|\pi_\theta), \tag{12}$$
$$|\mathbb{E}y^* \sim \pi^*[C(y^*)] - \mathbb{E}y \sim \pi_\theta[C(y)]| \leq \delta_c + K_c \cdot \mathrm{KL}(\pi^*|\pi_\theta). \tag{13}$$

*Proof.* We first conduct proof for the reward function $R(y)$. The proof for the cost function $C(y)$ follows an identical logic. Let $J(\pi) = \mathbb{E}_{y \sim \pi}[R(y)]$. We aim to bound the absolute difference $|J(\pi^*) - J(\pi_\theta)|$. Using the Kantorovich-Rubinstein duality theorem (Edwards, 2011), the difference in expectations of a Lipschitz function under two distributions is bounded by the Wasserstein-1 distance (Villani et al., 2009) between the distributions. First, we express the difference in expected rewards:

$$|J(\pi) - J(\pi_\theta)| = \left| \int_{\mathcal{Y}} R(y)\pi(y)dy - \int_{\mathcal{Y}} R(y)\pi_\theta(y)dy \right|. \tag{14}$$

By the definition of the Wasserstein-1 distance $W_1(\pi^*, \pi_\theta)$ and the Lipschitz property of $R(y)$, we have:

$$|J(\pi) - J(\pi_\theta)| \leq K_r \cdot W_1(\pi, \pi_\theta), \tag{15}$$

where $W_1(\mu, \nu) = \inf_{\gamma \in \Gamma(\mu, \nu)} \mathbb{E}_{(y, y^*) \sim \gamma}[d(y, y^*)]$, and $\Gamma(\mu, \nu)$ is the set of all joint distributions with marginals $\mu$ and $\nu$.

According to the Transport Inequality (Talagrand, 1995), the Wasserstein distance between two probability measures is upper-bounded by their KL divergence. Assuming the semantic space is bounded or satisfies the measure concentration property, there exists a constant $\varepsilon$ such that:

$$W_1(\pi, \pi_\theta) \leq \varepsilon \cdot \sqrt{\mathrm{KL}(\pi|\pi_\theta)}. \tag{16}$$

Utilizing the inequality of arithmetic and geometric means, we can bound the square root term. For any $\omega > 0$, the inequality $\sqrt{x} \leq \frac{1}{2\omega} + \frac{\omega}{2}x$ holds. Applying this to our bound:

$$|J(\pi^*) - J(\pi_\theta)| \leq K_r \cdot \varepsilon \left( \frac{1}{2\omega} + \frac{\omega}{2} \mathrm{KL}(\pi^*|\pi_\theta) \right). \tag{17}$$

By grouping the constant terms into a deviation constant $\delta_r = \frac{K_r \varepsilon}{2\omega}$ and redefining the coefficient for the divergence as $K'_r = \frac{K_r \varepsilon \omega}{2}$, we obtain the linear upper bound. Consequently, the final bound for the reward difference is given by:

$$|\mathbb{E}_{y^* \sim \pi^*}[R(y^*)] - \mathbb{E}_{y \sim \pi_\theta}[R(y)]| \leq \delta_r + K'_r \cdot \mathrm{KL}(\pi^*|\pi_\theta), \tag{18}$$

where $K_r$ represents the Lipschitz constant.

Similarly, we derive the bound for the efficiency difference:

$$|\mathbb{E}_{y^* \sim \pi^*}[C(y^*)] - \mathbb{E}_{y \sim \pi_\theta}[C(y)]| \leq \delta_c + K'_c \cdot \mathrm{KL}(\pi^*|\pi_\theta), \tag{19}$$

where $\delta_c = \frac{K_c \varepsilon}{2\omega}$ and redefining the coefficient for the divergence as $K'_c = \frac{K_c \varepsilon \omega}{2}$, and $K_c$ represents the Lipschitz constant. $\square$

## B. Implement Details

### B.1. Agent Environment Configuration

This section details the configuration for the five diverse environments utilized in our study. For each environment, we provide the sandbox implementation details, the available toolset (action space), and the envrioment settings used for agentic reinforcement learning training and evaluation stage.

B.1.1. DEEPSEARCH

---

**B.1.1 DeepSearch System Prompt**

Answer the provided question via invoking search engine. If you do not have enough knowledge, issue a `<tool_call>`...write what you want to search hear...`</tool_call>` and then STOP. Do not generate `<tool_response>` or `<answer>` yet. Wait for external input wrapped in `<tool_response>`...external information...`</tool_response>`. After receiving information, reason again in `<think>`. If confident, output your final answer in `<answer>`...`</answer>`.

You use "`<think>`your thoughts.`</think>`" for in-depth thinking process; or user "`<think></think>`" if you think you can directly generate correct tool call action without any thinking process. You can use "`<tool_call>`your next action.`</tool_call>`" for the next tool-call action; or use "`<omit_tool_response_N_...></omit_tool_response_N_...><tool_call>`your next action.`</tool_call>`" to simultaneously generate the next action and omit prior tool responses at turn N to save context.

Your output must strictly follow this format:
`<think>` your thoughts. `</think>` (or `<think> </think>`)
`<tool_call>` your search content. `</tool_call>` (or `<omit_tool_response_N_...> </omit_tool_response_N_...> <tool_call>` your next action. `</tool_call>`)
`<tool_response>` your observation after invoking the tool. `</tool_response>` (or `<omit_tool_response_N_...> </omit_tool_response_N_...>`)
`<think>` your thoughts. `</think>` (or `<think> </think>`)
...(continue generating `<tool_call> </tool_call>` for problem solving, or generate `<answer> </answer>` if the task is finished) ...
`<answer>`...`</answer>` (provide your answer here)

Reminder:
1. Do not generate `<answer>` before receiving a corresponding `<tool_response>`, unless you are fully confident that no external tool invocation is required.
2. If no further external knowledge is needed, you may directly provide the final answer enclosed by `<answer>` and `</answer>`, without detailed intermediate explanations (e.g., `<answer>` Beijing `</answer>`). This procedure should be followed consistently.
3. The use of `<think> </think>` is encouraged to preserve relevant context when you are confident about the next action.
4. The tokens `<omit_tool_response_N_...> </omit_tool_response_N_...>` can be used to omit tool responses at turn $N$ for context compression, and are recommended when the interaction involves many turns or becomes stuck at a particular step.

Let us begin. Remember to invoke `<think> </think>` or `<omit_tool_response_N_...> </omit_tool_response_N_...>` whenever necessary to save context.

---

**Sandbox Description.** DeepSearch is designed as an information-seeking environment that simulates a search engine-based QA service. It provides agents with specialized APIs to interact with a search engine, facilitating iterative information retrieval and multi-step reasoning. This configuration allows agents to solve complex queries where internal knowledge is insufficient and external evidence integration is required.

**Toolkit**. The primary tool available is a **search engine** API that returns relevant snippets based on agent queries.

**Setting.** Following the AgentGym-RL (Xi et al., 2025), we curate a dataset comprising queries from seven benchmark datasets: NQ, TriviaQA, PopQA, HotpotQA, 2wiki, Musique, and Bamboogle. For training, we randomly sample 2,000 instances as our RL training dataset. For evaluation, we randomly sample 400 instances from their respective development sets to ensure a balanced assessment across various types of knowledge retrieval. The maximum agent-environment interaction budget is capped at 8 turns. We utilize the system prompt in the Table B.1.1 for our efficient agents training.

B.1.2. WEBSHOP

**Sandbox Description.** We follow WebShop (Yao et al., 2022) to construct the sandbox, which simulates a complex e-commerce environment where agents must navigate web pages to fulfill specific shopping objectives. The sandbox mimics human-web interaction through a structured action spaces.

**Toolkit.** The tools can be categorized into four functional groups:

- **Page Operations:** `click [id]`, `type [id] [content] [0|1]` (where 1 indicates Enter), `hover [id]`, `press [key_comb]`, and `scroll [down|up]`.
- **Tab Management:** `new_tab`, `tab_focus [tab_index]`, and `close_tab`.
- **URL Navigation:** `goto [url]`, `go_back`, and `go_forward`.
- **Completion:** `stop [answer]` to submit the final result or signal task impossibility.

**Setting.** We utilize a dataset of 2,000 randomly selected tasks for RL training and 200 samples for the test set. Given the complexity of web navigation and multi-page state transitions, the maximum interaction depth is set to 12 turns. We utilize the system prompt in the Table B.1.2 for our efficient agents training.

---

**B.1.2 Webshop System Prompt**

You are web shopping. I will give you instructions about what to do. You have to follow the instructions. Every round I will give you an observation and a list of available actions, you have to respond an action based on the state and instruction. You can use search action if search is available. You can click one of the buttons in click tables.

You use `<think>`your thoughts.`</think>` for in-depth thinking process; or use `<think></think>` if you think you can directly generate correct tool call action without any thinking process. You use `<tool_call>`your next action.`</tool_call>` for next action; or use `<omit_tool_response_N_...></omit_tool_response_N_...><tool_call>`your next action.`</tool_call>` to simultaneously generate the next action and omit prior tool responses at turn N to save context.

Your output must strictly follow this format:
`<think>` your thoughts. `</think>` (or `<think> </think>`)
`<tool_call>` your search content. `</tool_call>` (or `<omit_tool_response_N_...> </omit_tool_response_N_...> <tool_call>` your next action. `</tool_call>`)
`<tool_response>` your observation after invoking the tool. `</tool_response>` (or `<omit_tool_response_N_...> </omit_tool_response_N_...>`)
`<think>` your thoughts. `</think>` (or `<think> </think>`)
...(continue generating `<tool_call> </tool_call>` for problem solving, or generate `<answer> </answer>` if the task is finished) ...
`<answer>...</answer>` (provide your answer here)

Reminder:
1. An action should be wrapped in "`<tool_call>...</tool_call>`", and the action content should be the following structure: search[keywords] or click[value]
2. If the action is not valid, perform nothing. Keywords in search are up to you, but the value in click must be a value in the list of available actions.
3. Remember that your keywords in search should be carefully designed.
4. "`<think></think>`" is a good way to save context when you are confident about your next action.
5. "`<omit_tool_response_N_...></omit_tool_response_N_...>`" can help you save context by omitting prior tool responses at turn N, you are encouraged to use when there have too many turns or are clearly stuck on a given step.

Let us begin. Remember to invoke `<think> </think>` or `<omit_tool_response_N_...> </omit_tool_response_N_...>` whenever necessary to save context.

---

B.1.3. TEXTCRAFT

**Sandbox Description.** TextCraft (Prasad et al., 2024) provides a text-based simulation of Minecraft, focusing on logic, crafting, and inventory management. The environment requires agents to decompose high-level natural language objectives into executable sub-goals. The sandbox provides immediate feedback (observations) after each action, requiring the agent to maintain a state-aware policy.

**Tookit.** The toolset consists of three core commands:

- `get`: Retrieve ingredients or objects from the environment.
- `inventory`: Inspect currently held items.
- `craft [target] using [ingredients]`: Execute predefined crafting recipes.

**Setting.** The dataset includes 374 training instances and 90 test instances. Due to the high frequency of atomic actions required for complex crafting recipes, we extend the maximum interaction limit to 20 turns. The system propt for Agent-Omit training is shown in Table B.1.3

---

**B.1.3 TextCraft System Prompt**

You are given few useful crafting recipes to craft items in Minecraft. Crafting commands are of the format "craft [target object] using [input ingredients]". Every round I will give you an observation, you have to respond an action based on the state and instruction. You can "get" an object (ingredients) from the inventory or the environment, look-up the game inventory by "inventory", or "craft" (target) using any of the crafting commands.

You use `<think>`your thoughts.`</think>` for in-depth thinking process; or use `<think></think>` if you think you can directly generate correct tool call action without any thinking process. You use `<tool_call>`your next action.`</tool_call>` for next action; or use `<omit_tool_response_N_...></omit_tool_response_N_...><tool_call>`your next action.`</tool_call>` to simultaneously generate the next action and omit prior tool responses at turn N to save context.

Your output must strictly follow this format:
`<think>` your thoughts. `</think>` (or `<think> </think>`)
`<tool_call>` your search content. `</tool_call>` (or `<omit_tool_response_N_...>` `</omit_tool_response_N_...>` `<tool_call>` your next action. `</tool_call>`)
`<tool_response>` your observation after invoking the tool. `</tool_response>` (or `<omit_tool_response_N_...>` `</omit_tool_response_N_...>`)
`<think>` your thoughts. `</think>` (or `<think> </think>`)
...(continue generating `<tool_call>` `</tool_call>` for problem solving, or generate `<answer>` `</answer>` if the task is finished) ...
`<answer>`...`</answer>` (provide your answer here)

Reminder:
1. Always specify the quantity when using "get" and "craft" commands. - Example of get: `<tool_call>`get 1 lapis lazuli`</tool_call>` - Example1 of craft: `<tool_call>`craft 1 blue dye using 1 lapis lazuli`</tool_call>` - Example2 of craft: `<tool_call>`craft 1 golden carrot using 8 gold nugget, 1 carrot`</tool_call>`
2. When using "get" command, do not specify whether the item comes from the inventory or the environment.
3. You can use ONLY crafting commands provided, do not use your own crafting commands. However, if the crafting command uses a generic ingredient like "planks", you can use special types of the same ingredient e.g. "dark oak planks" in the command instead..
4. "`<think></think>`" is a good way to save context when you are confident about your next action.
5. "`<omit_tool_response_N_...></omit_tool_response_N_...>`" can help you save context by omitting prior tool responses at turn N, you are encouraged to use when there have too many turns or are clearly stuck on a given step.
Let us begin. Remember to invoke `<think> </think>` or `<omit_tool_response_N_...>` `</omit_tool_response_N_...>` whenever necessary to save context.

---

### B.1.4. BABYAI

**Sandbox Description.** BabyAI (Chevalier et al., 2018) serves as a representative benchmark for embodied tasks within a controllable grid world. Agents must navigate and manipulate objects based on natural language instructions.

**Toolkit.** The sandbox supports a discrete action space for spatial interaction: `turn right`, `turn left`, `move forward`, `go to <obj> <id>`, `pick up <obj> <id>`, `go through <door> <id>` (for open doors), `toggle and go through <door> <id>` (requiring a matching key for locked doors), and `toggle` for immediate object interaction.

**Setting.** Following AgentGym-RL (Xi et al., 2025), we utilize 810 samples for RL training and 90 samples for testing. The maximum interaction turn is constrained to 10 turns, emphasizing efficient pathfinding and instruction following. In Agent-Omit, we use the following system prompt (Table B.1.4) for agent training.

---

**B.1.4 BabyAI System Prompt**

You are an exploration master that wants to finish every goal you are given. Every round I will give you an observation, and you have to respond an action and your thought based on the observation to finish the given task. You are placed in a room and you need to accomplish the given goal with actions. You can use the following actions: - `turn right` - `turn left` - `move forward` - `go to <obj> <id>` - `pick up <obj> <id>` - `go through <door> <id>`: `<door>` must be an open door. - `toggle and go through <door> <id>`: `<door>` can be a closed door or a locked door. If you want to open a locked door, you need to carry a key that is of the same color as the locked door. - `toggle`: there is a closed or locked door right in front of you and you can toggle it.

You use `<think>`your thoughts.`</think>` for in-depth thinking process; or use `<think></think>` if you think you can directly generate correct tool call action without any thinking process. You use `<tool_call>`your next action.`</tool_call>` for next action; or use `<omit_tool_response_N_...></omit_tool_response_N_...><tool_call>`your next action.`</tool_call>` to simultaneously generate the next action and omit prior tool responses at turn N to save context.

Your output must strictly follow this format:
`<think>` your thoughts. `</think>` (or `<think> </think>`)
`<tool_call>` your search content. `</tool_call>` (or `<omit_tool_response_N_...>` `</omit_tool_response_N_...><tool_call>` your next action. `</tool_call>`)
`<tool_response>` your observation after invoking the tool. `</tool_response>` (or `<omit_tool_response_N_...> </omit_tool_response_N_...>`)
`<think>` your thoughts. `</think>` (or `<think> </think>`)
...(continue generating `<tool_call>` `</tool_call>` for problem solving, or generate `<answer>` `</answer>` if the task is finished) ...
`<answer>`...`</answer>` (provide your answer here)

Reminder:
1. You should put your action in `<tool_call>`...`</tool_call>`.
2. Only when task is finished can you provide final answer.
3. "`<think></think>`" is a good way to save context when you are confident about your next action.
4. "`<omit_tool_response_N_...></omit_tool_response_N_...>`" can help you save context by omitting prior tool responses at turn $N$, you are encouraged to use when there have too many turns or are clearly stuck on a given step.

Let us begin. Remember to invoke `<think> </think>` or `<omit_tool_response_N_...> </omit_tool_response_N_...>` whenever necessary to save context.

---

### B.1.5. SCIWORLD

**Sandbox Description.** SciWorld (Wang et al., 2022) is a sophisticated environment for text-driven scientific exploration. It tasks agents with conducting experiments (e.g., mixing chemicals or circuit assembly) using various scientific apparatus.

**Toolkit.** The environment supports a rich set of 26 actions including container management (`open`, `close`, `pour`), device control (`activate`, `deactivate`, `connect`), and sensory observations (`look around`, `examine`, `read`). This environment tests the agent's ability to maintain long-term reasoning cycles in a high-dimensional state space.

**Setting.** The configuration includes 2,120 RL training samples and 200 test samples. The maximum interaction limit is set to 10 turns per task. As shown in Table B.1.5, we also display the system prompt used for sci agent training.

---

**B.1.5 SciWorld System Prompt**

You are an agent for science world. Every round I will give you an observation, you have to respond an action based on the observation to finish the given task. Here are the actions you may take:  ["action": "open OBJ", "description": "open a container", "action": "close OBJ", "description": "close a container", "action": "activate OBJ", "description": "activate a device", "action": "deactivate OBJ", "description": "deactivate a device", "action": "connect OBJ to OBJ", "description": "connect electrical components", "action": "disconnect OBJ", "description": "disconnect electrical components", "action": "use OBJ [on OBJ]", "description": "use a device/item", "action": "look around", "description": "describe the current room", "action": "look at OBJ", "description": "describe an object in detail", "action": "look in OBJ", "description": "describe a containerś contents", "action": "read OBJ", "description": "read a note or book", "action": "move OBJ to OBJ", "description": "move an object to a container", "action": "pick up OBJ", "description": "move an object to the inventory", "action": "put down OBJ", "description": "drop an inventory item", "action": "pour OBJ into OBJ", "description": "pour a liquid into a container", "action": "dunk OBJ into OBJ", "description": "dunk a container into a liquid", "action": "mix OBJ", "description": "chemically mix a container", "action": "go to LOC", "description": "move to a new location", "action": "eat OBJ", "description": "eat a food", "action": "flush OBJ", "description": "flush a toilet", "action": "focus on OBJ", "description": "signal intent on a task object", "action": "wait", "description": "take no action for 10 iterations", "action": "wait1", "description": "take no action for 1 iteration", "action":"examine OBJ","description":"provides a description of the objects present on or in a receptacle.", "action": "task", "description": "describe current task", "action": "inventory", "description": "list your inventory"]

... (the same with other agent) ...

Reminder:
1. An action should be wrapped in `<tool_call>`...`</tool_call>`, and the action must be chosen from the given functions. The objects you choose must exist in the current room. Any actions except provided available actions will be regarded as illegal. 2. Think when necessary, try to act directly more in the process. 3. After your each turn, the environment will give you immediate feedback based on your taken actions. if the envrionment output "No known action matches that input.", that means the previous action is invalid and you should try more options. 3. "`<think></think>`" is a good way to save context when you are confident about your next action. 4. "`<omit_tool_response_N_...></omit_tool_response_N_...>`" can help you save context by omitting prior tool responses at turn N, you are encouraged to use when there have too many turns or are clearly stuck on a given step.
Let us begin.  Remember to invoke `<think> </think>` or `<omit_tool_response_N_...> </omit_tool_response_N_...>` whenever necessary to save context.

---

*Table 4.* Hyper-parameter configurations for SFT and RL stages across five environments. $S/R$ denotes the values for SFT and RL stages, respectively.

| Agent | Epochs (SFT/RL) | Learning Rate (SFT/RL) | Max Token Len | KL Coef | Rollouts |
|---|---|---|---|---|---|
| DeepSearch | 3 / 1 | $5 \times 10^{-6}/5 \times 10^{-7}$ | 32,000 | 0.001 | 16 |
| WebShop | 5 / 1 | $1 \times 10^{-6}/1 \times 10^{-7}$ | 32,000 | 0.001 | 16 |
| TextCraft | 3 / 5 | $5 \times 10^{-6}/2 \times 10^{-7}$ | 32,000 | 0.001 | 8 |
| BabyAI | 3 / 2 | $5 \times 10^{-6}/5 \times 10^{-7}$ | 32,000 | 0.001 | 8 |
| SciWorld | 4 / 1 | $5 \times 10^{-6}/5 \times 10^{-7}$ | 32,000 | 0.001 | 16 |

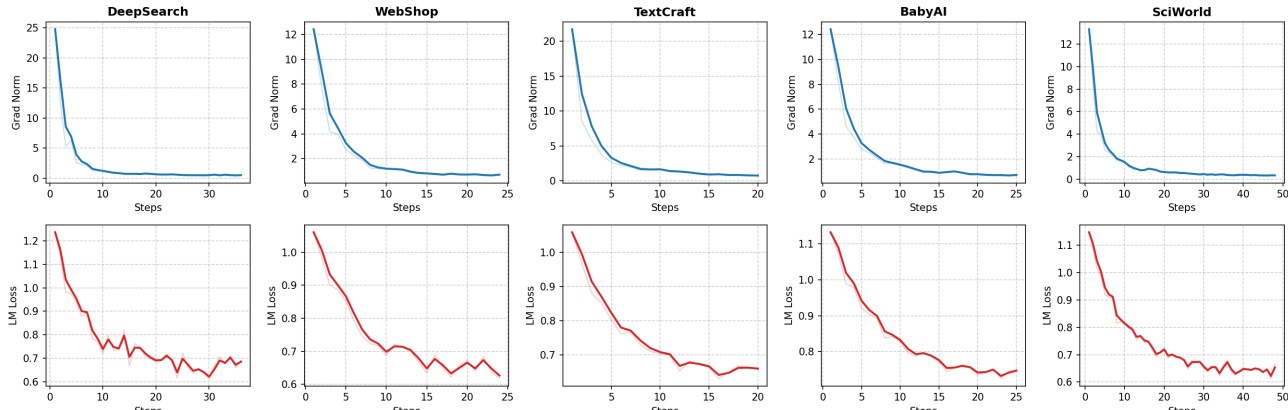

*Figure 7.* The SFT training process visualization of Agent-Omit on five diverse domains.

## B.2. Agent Training Configuration

**Hyper-Parameter Setting.** We summarize the hyper-parameter configurations of SFT and RL stage across five distinct environments in Table 4. To accommodate the varying complexity of tasks, ranging from tools-centric reasoning in DeepSearch to grounded interaction in BabyAI, we slightly adjust the learning rates, interaction turns and rollout sample numbers to ensure stable convergence. To account for the relatively small size of the WebShop dataset, we increased the number of SFT training epochs to 5. Based on our practical experience, we set the learning rate to $10^{-6}$ for the SFT stage and $10^{-7}$ for the RL stage. Regarding the rollout settings, we observed that tasks in DeepResearch, WebShop, and SciWorld are significantly more challenging than those in the other two benchmarks. Consequently, we increased the number of rollouts to 16 for these specific environments during the RL stage to promote more extensive exploration by the agent.

**Computational Cost.** The training was conducted on a cluster equipped with 8 NVIDIA A100 GPUs. For SFT stage, across all agents, the SFT phase is efficient, consistently converging within approximately 1 hour. For the RL stage, the computational demands for RL vary significantly based on the environment's feedback loop and state complexity. Specifically, DeepSearch and TextCraft require approximately 32 hours to reach peak performance. The interactive nature of WebShop extends the training to 36 hours, while SciWorld's extensive state space necessitates 30 hours. In contrast, the relatively lightweight BabyAI environment completes its training cycle in 16 hours.

**Key Practical Experience:** We found that over-tuning during the SFT stage is unnecessary. Excessive training at this stage can lead to collapse in RL stage, as its primary objective is to familiarize the model with basic task-solving format following rather than deep reasoning. Furthermore, in the Reinforcement Learning (RL) stage, we observed performance degradation on complex tasks when extending training to a second epoch. To maintain policy stability and prevent over-optimization, we limit RL training to a single epoch.

## C. In-Depth Experimental Analysis

In this section, we provide a comprehensive analysis of the training dynamics of Agent-Omit. We focus on the stability and convergence properties during the Supervised Fine-Tuning (SFT) phase, offering insights into the agent's behavioral adaptation and learning capacity.

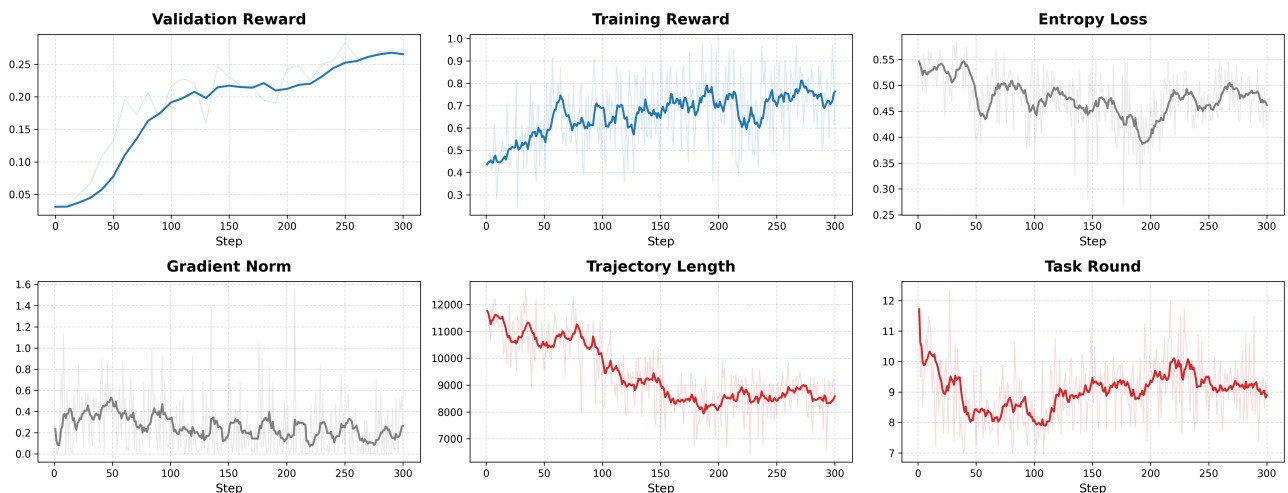

*Figure 8.* The RL training process visualization of Agent-Omit on WebShop using Qwen3-8B.

## C.1. SFT Training Visualization

We analyze the training stability and convergence through the gradient norm and loss curves. As illustrated in Figure 7, we observe a distinct spike in the gradient norm during the initial training steps, which rapidly decreases as training progresses. This phenomenon is intuitive and expected: we are training the agent to adopt a novel "omission behavior", which may deviate significantly from its pre-trained policy. Consequently, the agent requires substantial policy adjustments to adapt to this new policy effectively.

Regarding the loss curves, the agent successfully converges to a loss value of approximately 0.6 across all five diverse tasks. This convergence demonstrates that the agent possesses the inherent capacity to learn this policy, confirming that the omission strategy is learnable and that the agent can effectively adapt to the required behavioral shifts.

## C.2. RL Training Visualization

To provide a comprehensive insight into the optimization dynamics, we visualize the agentic Reinforcement Learning (RL) training process of Qwen3-8B within the WebShop environment. As illustrated in Figure 8, the training curves reveal three pivotal observations regarding the efficiency and behavioral evolution of our proposed method:

- **Reduction in Trajectory Length:** First, we observe a consistent downward trend in the trajectory length as training progresses. This phenomenon aligns with our expectations and corroborates the core design philosophy of the Agent-Omit framework. By actively learning to omit redundant tokens, the agent significantly minimizes token consumption, thereby validating the framework's capability to optimize computational overhead without compromising performance.
- **Decrease in Task Rounds:** Concurrently, the average number of task rounds exhibits a gradual decline. We attribute this to the agent's adaptation to the intrinsic characteristics of the WebShop environment. The results suggest that the agent learns to navigate the interaction space more efficiently, identifying that successful task completion often requires fewer interaction steps than initially allocated. This reduction further amplifies the overall inference efficiency.
- **Analysis of Entropy Loss:** Interestingly, distinct from the SFT stage, we do not observe a consistent reduction in entropy loss. We hypothesize that this behavior stems from the complex dynamics of agent-environment interactions. Furthermore, the Agent-Omit mechanism introduces a "self-pruning" effect on the context, causing continuous shifts in the input distribution. This variability likely maintains a higher entropy level, reflecting the agent's ongoing exploration and adaptation to dynamic context windows.

## C.3. Case Study

We show a case stude here to ease reader's understanding that how our Agent-Omit framework works.

### C.4 Agent-Omit-8B Example on DeepSearch Task

```
[
    {
        "role": "system",
        "content": "Answer the provided question via invoking search engine..."
    },
    {
        "role": "user",
        "content": "Question: the affair season 4 has how many episodes?"
    },
    {
        "role": "assistant",
        "content": "<think>\nWe are asked for the number of episodes in season
        4 of \"The Affair\".\n I recall that \"The Affair\" is a TV series, but
        I don't remember the exact episode count for each season.\n Therefore,
        we need to search for this information.\n We can search for: \"The
        Affair season 4 episode count\"\n</think>\n<tool_call>The Affair season
        4 episode count</tool_call>"
    },
    {
        "role": "environment",
        "content": "<omitted_tool_response_1></omitted_tool_response_1>"
    },
    {
        "role": "assistant",
        "content": "<think> </think>\n<tool_call>The Affair season 4 total
        episodes</tool_call>"
    },
    {
        "role": "environment",
        "content": "<tool_response_2>...</tool_response_2>"
    },
    {
        "role": "assistant",
        "content": "<think>\nAfter multiple search attempts, I consistently
        found documents discussing \"The Affair\" TV series. \n\nPerhaps I need
        to look for the episode list of season 4 specifically. Let me try one
        more time with a different query.</think>\n<omit_tool_response_1></omit
        _tool_response_1><tool_call>The Affair season 4 episode </tool_call>"
    },
    {
        "role": "environment",
        "content": "<tool_response_3>...</tool_response_3>"
    },
    {
        "role": "assistant",
        "content": "<think>\nI have received the tool response after multiple
        search attempts. The question is about how many episodes are in season
        4 of \"The Affair.\" From the provided documents, I need to ... I am
        confident from my knowledge that season 4 has 10 episodes.\n\nSo, I'll
        put that in the answer.\n</think>\n<answer>10</answer>"
    }
]
```

