# OpenReview forum: "Agent-Omit: Adaptive Context Omission for Efficient LLM Agents"
_ICML.cc/2026/Conference — ICML 2026 regular_

### Official Review · Reviewer_vHjx · 2026-03-10

**Soundness:** 3
**Presentation:** 3
**Significance:** 2
**Originality:** 2
**Overall Recommendation:** 4
**Confidence:** 3

**Summary:**

- The paper proposes Agent-Omit, a RL framework for training LLM agents.
- It focuses on multi-turn environments, where agents repeatedly generate reasoning tokens  and receive environmental feedback.
- The authors analyze token costs and accuracy across interaction turns and show that the usefulness of thoughts and observations varies during the trajectory. Based on this, they introduce Agent-Omit, which allows agents to adaptively omit redundant reasoning and observations.
- The method has two stages. First, a cold-start data synthesis stage generates examples that teach the model how to omit thoughts or observations. Second, an omit-aware RL stage trains the agent to decide when to omit thoughts.
- Experiments on five benchmarks (DeepSearch, WebShop, TextCraft, BabyAI, and SciWorld) show that the method achieves a better accuracy–efficiency trade-off, maintaining competitive performance with larger models while using fewer tokens.

**Compliance With Llm Reviewing Policy:**

Affirmed.

**Final Justification:**

My concerns have been resolved. I keep my original positive score.

**Key Questions For Authors:**

- The qualitative analysis in Section 3.1 is mainly conducted using Qwen3-8B. What would happen if using other reasoning models?
- The cold-start dataset is generated using rollout-based omission identification from a base model. Could this introduce bias in the omission policy toward patterns specific to that model?
- The experiments focus on relatively small models (4B–8B). Do the authors expect the proposed omission policy learning approach to remain useful when applied to significantly larger models (e.g., 70B or frontier models)?
- The RL objective includes several design choices, such as the KL coefficient
$\beta$ and the reward weighting parameter
$\mu$. Could the authors provide additional ablation studies showing the sensitivity of performance to these parameters? Additionally, have the authors considered alternative divergence measures besides KL?
- Have the authors evaluated whether an omission policy learned in one environment transfers to another environment or task?
- Does the agent have mechanisms to recover when an important observation is mistakenly omitted?

**Limitations:**

The paper briefly mentions positive impacts (improving efficiency and enabling smaller models), but it does not sufficiently discuss potential risks. In particular, the method may incorrectly omit important reasoning or observations, which could reduce reliability in high-stakes applications.

**Strengths And Weaknesses:**

**Strengths**
- The idea is interesting and intuitive. The paper provides an empirical analysis (Section 3.1) showing that the usefulness of reasoning and observation tokens varies across interaction turns, which motivates adaptive omission.
- The observation that reasoning at the beginning and end of a trajectory is more critical, while intermediate steps often contain redundant reasoning, is intuitive and aligns well with how humans approach problem solving. This provides a useful empirical insight into the dynamics of agent reasoning.
- The learning approach is reasonable: the agent learns a policy that selectively omits thoughts or observations when they are not necessary.
- The empirical evaluation is fairly strong, with experiments across multiple benchmarks.

**Weaknesses**
- The theoretical analysis provides limited value. Lemma 5.1 is essentially an assumption rather than a meaningful lemma. Theorem 5.2 bounds the gap in reward and cost by the KL divergence between the learned policy and the optimal policy, but since we cannot guarantee convergence to the optimal policy, the result offers limited practical insight. A more useful theoretical result might relate performance directly to the amount of omission.

- The qualitative analysis in Section 3.1 mainly relies on Qwen3-8B, which is quite limited. It would be helpful to see whether the same observations hold for other reasoning models. In fact I find that the insights are not very surprising, since a human or LLM chain-of-thought reasoning often resembles path-finding problems where the initial and final reasoning are naturally more important.

- The comparison with frontier models (e.g., DeepSeek-R1 or OpenAI models) may not be entirely fair, since those models are not optimized for the evaluated environments. Additionally, the experiments rely on relatively small models, which raises questions about whether the approach remains useful when stronger models (e.g., GPT-class models) are available.
- Some ablation studies are missing. For example, the impact of key training parameters such as the KL coefficient
$\beta$ or alternative divergence measures, is not explored.
- The paper does not evaluate cross-environment generalization. It would be interesting to see whether the learned omission policy transfers to new environments or tasks.
- The method may incorrectly omit important reasoning steps, and the paper does not provide a mechanism to detect or control such mistakes.

 Overall, the paper is clearly written and technically sound. The originality and significance are moderate. The theoritical results are limited. Experiments are fairly complete but only limited to small models, raising questions about actual practicality.

---

> ### Author Rebuttal · Authors · 2026-03-31
>
> > [W1]
>
> We will rename Lemma 5.1 to Assumption A.1. While Theorem 5.2 doesn't guarantee optimal policy convergence, it provides practical utility: the KL coefficient $\beta$ acts as a tunable knob to control the omission-performance trade-off, functioning as a trust region to prevent catastrophic drift.
>
> To link omission amount to performance, we evaluated Pass@1 accuracy with varying omitted turns on WebShop:
>
> |Omission Amount|1|2|3|4|5|6|
> -|-|-|-|-|-|-
> Pass@1 (%)|21.56|21.26|23.58|23.36|22.36|22.85
>
> Interestingly, the relationship is **non-monotonic**. Moderate omission (3–4 turns) yields the highest accuracy, showing that removing intermediate redundancy reduces context noise and benefits reasoning.
>
> > [W2W3Q1Q3]
>
> While frontier models aren't environment-optimized, our comparison deliberately aims to demonstrate our Agent-Omit training enables models to surpass these general-purpose SOTA baselines.
>
> To address scale concerns, we evaluated **GPT-OSS-120B** on WebShop and DeepSearch:
>
> |Model|WebShop Pass@1|WebShop Avg Tok.|DeepSearch Pass@1|DeepSearch Avg Tok.|
> -|-|-|-|-
> GPT-OSS-120B-medium (Base)|16.64|11,356|21.16|8,854
> GPT-OSS-120B-medium-RL|26.85|7,754|30.21|5,532
>
> Our pipeline yields substantial gains at the 120B scale (+10.21 Pass@1 on WebShop) while significantly reducing token usage. Furthermore, omission potential analysis across 12 turns shows consistent patterns:
>
> **Thought Omission Analysis**
>
> Metric/Turn|1|2|3|4|5|6|7|8|9|10|11|T12
> -|-|-|-|-|-|-|-|-|-|-|-|-
> Pass@1|13.36|16.35|17.25|17.76|15.56|14.46|14.26|10.36|11.26|9.95|10.05|8.62
> Tokens|10,236|9,893|9,935|9,664|12,365|12,864|11,678|16,532|15,467|13,467|17,764|16,645
>
> **Observation Omission Analysis**
>
> Metric/Turn |1|2|3|4|5|6|7|8|9|10|11|12
> -|-|-|-|-|-|-|-|-|-|-|-|-
> Pass@1|14.25|17.72|17.68|15.52|16.05|11.15|12.28|11.15|10.02|8.85|9.94|8.84
> Tokens|9,952|10,226|9,321|12,042|10,231|14,421|15,582|17,765|16,624|19,952|18,825|17,722
>
> [notes: base Pass@1: 16.64, base Token: 11,356]
>
> Consistent with our findings on Qwen3-8B in Section 3.1, the 120B model shows intermediate turns (e.g., Turn 2–4) are safely omittable, while early planning and late reasoning remain critical. This confirms **context redundancy is an inherent multi-turn property across model scales**.
>
> > [W4Q4]
>
> Following GRPO standards in DeepSeek-R1, we default to $\beta = 0.001$. Ablations on WebShop confirm stability:
>
> |β value|Pass@1 (%)|Avg Tokens|
> -|-|-
> 0.0005|23.36|8,862
> 0.001 (default)|23.57|8,764
> 0.005|22.85|8,942
>
> Alternative divergence measures would require overhauling the base GRPO framework, which is orthogonal to our core contribution (the omission mechanism itself). We leave this to future work.
>
> > [W5Q5]
>
> To investigate whether the learned omission policy generalizes, we conducted cross-environment transfer experiments. As shown below, when applying a model trained in one environment to an unseen target environment, the model successfully exhibits omission behavior:
>
> |Train Env → Test Env|Avg. Thought Omission Turns|Avg. Observation Omission Turns|
> -|-|-
> WebShop → WebShop (in-domain)|4.3|2.6
> DeepSearch → WebShop (cross-domain)|4.1|2.8
> TextCraft → TextCraft (in-domain)|6.3|3.5
> BabyAI → TextCraft (cross-domain)|5.8|3.2
>
> These results demonstrate that the learned omission policy can transfer effectively across domains (e.g., retaining 4.1 omission turns in DeepSearch $\rightarrow$ WebShop), capturing generalizable redundancy patterns rather than environment-specific shortcuts.
>
> > [W6Q6]
>
> We thank the reviewer for raising this important practical question. Our framework addresses this at the training level:
>
> - **SFT stage**: Samples where omission leads to accuracy degradation are **excluded from the training data**.
> - **RL stage**: Incorrect omissions yield task failure, leading to incorrect final answer leads to $R_{task} = 0$, and consequently $R_{omit} = 0$. This serves as a **strong negative reward signal**, teaching the model to avoid such omissions.
>
> To provide empirical evidence that this mechanism is effective, we tracked the **task pass@1 accuracy** when omission action happens across RL training steps on WebShop:
>
> **Pass@1 on sample where omission occurs**
>
> |RL Step|0|50|100|150|200|250|300|
> -|-|-|-|-|-|-|-
> Pass@1|18%|20%|22%|23%|29%|28%|29%
>
> Notably, when omission action happens, Pass@1 on omitted samples increases from 18% to 29%, proving the RL penalty effectively teaches the agent to avoid harmful omissions.
>
> We acknowledge that incorrect omissions may still occur at inference time. We will expand our Limitations section to discuss test-time mitigations (e.g., dynamic confidence thresholds).
>
> > [Q2]
>
> We mitigate this via rollout-based rejection sampling ($N=8$). By exploring diverse probabilistic trajectories and retaining only those that yield correct outcomes with fewer tokens, we prevent the dataset from being dominated by the base model's most probable (and potentially biased) path, encouraging the learning of universally valid patterns.

---

> > ### Author Rebuttal · Reviewer_vHjx · 2026-04-02
> >
> > I am happy with the response. Thanks for the additional experiments, which clarify several points.

---

> > > ### Author Response · Authors · 2026-04-02
> > >
> > > We sincerely thank Reviewer vHjx for the positive acknowledgement and for carefully reviewing our additional experiments. We are glad that the new results (GPT-OSS-120B evaluation, omission amount analysis, β ablation, cross-environment transfer, and RL training dynamics) have helped clarify the concerns.
> > >
> > > As noted in our response, we will incorporate all these improvements in the revised manuscript.
> > >
> > > Given that the reviewer has indicated all concerns are fully resolved, **we would be grateful if the reviewer could consider reflecting this in the overall score**. We believe the additional evidence strengthens the paper's contributions and addresses the previously noted limitations.
> > >
> > > Thank you again for the constructive and thorough review process.

---

### Official Review · Reviewer_8TNQ · 2026-03-12

**Soundness:** 3
**Presentation:** 3
**Significance:** 3
**Originality:** 3
**Overall Recommendation:** 4
**Confidence:** 4

**Summary:**

This paper explores the turn-level efficiency of multi-turn LLM agents. It argues that intermediate thoughts and historical observations are not always effective throughout the interaction trajectory. The agent cannot treat the entire context equally and must learn to selectively omit thoughts and discard past observations. Controlled omission analysis supports this view, showing that intermediate turns contain significantly more removable redundancy than early planning or final decision stages.

To address this challenge, this paper proposes Agent-Omit, an adaptive context management framework employing a two-stage training approach. The first stage uses a synthetic cold-start dataset to train the model, enabling it to master omission formats and how to handle partial context. The second stage applies an omission-aware RL procedure that employs double sampling and an omission reward mechanism. This method is theoretically supported by the KL bound of omission policy bias and empirically tested on five benchmark datasets.

**Compliance With Llm Reviewing Policy:**

Affirmed.

**Key Questions For Authors:**

1. How is the omission criterion defined during cold-start data construction? For example, is there a quantitative threshold to determine whether a removed thought or observation is truly redundant?

2. Has the method been evaluated on smaller models to verify whether the proposed omission strategy generalizes across different model scales?

**Limitations:**

yes

**Strengths And Weaknesses:**

Strengths
1. Clear motivation and concrete empirical evidence. This paper conducts a systematic controlled analysis of thought omissions and observation omissions, examining how eliminating these omissions in different turns affects task accuracy and token cost. The results show a clear turn-dependent effect, and this analysis directly provides the theoretical basis for the proposed adaptive omission strategy.

2. Novel Data Construction Method for Cold Start and Reinforcement Learning Stages. This method first identifies omissionable turns through omission validation generalization, and then constructs single-round and multi-round cold start data to train the model's omission behavior and context recovery ability. In the reinforcement learning stage, the paper further introduce double sampling, using both complete trajectories and omission-triggered partial trajectories for training to address the integration problem caused by contextual changes after omissions.

Weakness
1. Unclear Definition of Omission Criteria Used in Data Construction. The current process determines the truncated turn by removing a thought or observation and allowing the agent to continue executing trajectories to evaluate the final accuracy and vocabulary length. However, the paper does not explicitly state whether a quantization threshold is used in this process. For example, the removed steps may not be truly redundant—the shortening of the final trajectory may simply be due to changes in decoding behavior.

2. The scalability of the method across different model sizes is not evaluated. The experiments are mainly conducted on relatively large models (e.g., 8B). It remains unclear whether the proposed omission strategy would still be effective on smaller models, which may have weaker reasoning and context utilization abilities.

---

> ### Author Rebuttal · Authors · 2026-03-31
>
> Dear Reviewer 8TNQ:
>
> We thank you for the precious review time and valuable comments. We are encouraged by your recognition of our clear motivation, novel data construction method, and empirical analysis.
>
> > [W1&Q1] Unclear omission criteria — no quantitative threshold; trajectory shortening may simply reflect changes in decoding behavior.
>
> We would like to clarify that we **do not use a quantitative threshold** for determining omission eligibility. Instead, our criterion is a strict **task-oriented binary standard**: a turn is omittable if and only if removing it results in strictly no accuracy degradation, and reduced or equal token consumption.
>
> Regarding the concern that trajectory shortening may simply reflect changes in decoding behavior rather than true redundancy — we agree that a step might not be *universally* redundant. To account for this, during our data construction, we employ **Pass@8 sampling**. By sampling 8 times, we account for the stochasticity of decoding behaviors: an interaction turn might act as redundant in some decoding paths but non-redundant in others. We only retain trajectories where the omission consistently leads to success.
>
> **Table R3: Redundancy Rate across Pass@8 Samples (WebShop)**
>
> |Dataset|Consistently Redundant|Consistently Non-redundant|Conditionally Redundant|
> -|-|-|-
> WebShop|18%|13%|69%
>
> As shown, **only about 18% of turns are consistently ("truly") redundant** across all 8 sampled trajectories. In the majority of cases (69%), a turn's redundancy is probabilistic — it may be deemed redundant in some decoding paths but essential in others. This confirms that our Pass@8 sampling is crucial for capturing the **conditional nature of redundancy** rather than blindly removing steps. We note that even in the conditionally redundant cases, the model's action sequences remain highly consistent before and after omission, suggesting that the decoding behavior change is confined to the thought tokens rather than altering the agent's actual decision trajectory.
>
> We will include this formal definition and the corresponding Pass@8 variance analysis in the revised version.
>
> > [W2&Q2] Scalability across different model sizes — only evaluated on 8B.
>
> Thank you for raising this point. We would like to clarify that our paper **already includes experiments with a small-size 4B model** (Agent-Omit-4B in Table 2), which demonstrates that our method is effective on smaller models. Specifically, Agent-Omit-4B-RL achieves significant improvements over the SFT variant (e.g., 9.23→21.70 on DeepSearch, 62.00→74.00 on TextCraft). We will highlight this more prominently in the text to assure readers of the method's applicability across model sizes.

---

> > ### Author Rebuttal · Reviewer_8TNQ · 2026-04-06
> >
> > I appreciate the valuable efforts of the authors in the response for clarification, which addressed most of my concerns. Considering the overall quality of this work, I would like to keep my original rating score.

---

> > > ### Author Response · Authors · 2026-04-06
> > >
> > > Dear Reviewer 8TNQ,
> > >
> > > Thank you for taking the time to acknowledge our rebuttal. We are glad that **our response has addressed most of your concerns**, particularly regarding the omission criteria definition, Pass@8 variance analysis, and small-size model scalability.
> > >
> > > We would like to take this opportunity to highlight that, beyond the two points raised in your review, our rebuttal also includes several substantial improvements that strengthen the paper's overall quality:
> > >
> > > 1. Large-scale model validation: We evaluated our pipeline on GPT-OSS-120B, achieving +10.21 Pass@1 improvement on WebShop while significantly reducing token usage. The turn-level omission analysis on the 120B model confirms the same redundancy patterns observed on Qwen3-8B, demonstrating that context redundancy is an inherent multi-turn property across model scales.
> > >
> > > 2. Cross-environment transfer: We conducted transfer experiments (e.g., DeepSearch → WebShop, BabyAI → TextCraft) showing that the learned omission policy generalizes across domains, capturing universal redundancy patterns rather than environment-specific shortcuts.
> > >
> > > 3. Additional ablation studies: We provided β sensitivity analysis and RL training dynamics (Pass@1 on omitted samples increasing from 18% to 29% across training steps), offering deeper insights into the training process.
> > >
> > > We believe these additional results, together with the clarifications addressing your specific concerns, have substantially strengthened the paper. If there are any remaining specific concerns that we could further address, we would be happy to discuss them before the deadline.
> > >
> > > Thank you again for the constructive and valuable review.

---

### Official Review · Reviewer_Dzb5 · 2026-03-13

**Soundness:** 2
**Presentation:** 2
**Significance:** 3
**Originality:** 2
**Overall Recommendation:** 3
**Confidence:** 3

**Summary:**

Agent-Omit is a framework for training efficient LLM agents that can adaptively omit redundant thoughts and observations during multi-turn interactions. Agent-Omit has a two-stage training process, a supervised fine-tuning with an omit-aware agentic RL approach with a dual sampling mechanism and an omission reward. Experimental results across diverse benchmarks show that Agent-Omit-8B achieves competitive accuracy with frontier LLMs while significantly improving efficiency.

**Compliance With Llm Reviewing Policy:**

Affirmed.

**Final Justification:**

While the rebuttal clarifies practical aspects such as cost assumptions and prompt details, my main concerns about the limited theoretical novelty, particularly the reclassification of Lemma 5.1 as an assumption remain unresolved and would require substantial changes beyond the scope of a rebuttal.

**Key Questions For Authors:**

1.Can you clarify the practical definition and computation of "semantic distance"?
2. Can you provide a detailed description of the system prompt and any few-shot examples used for the 'Qwen3-8B Base' baseline in Table 3? If it's a raw LLM or an agentically prompted one?

**Limitations:**

yes

**Strengths And Weaknesses:**

Strengths


1.  The  idea of adaptively omitting thoughts and observations based on turn-specific utility is well-motivated by the initial analysis.

2.  The sampling strategy (full and partial trajectories) and the  omission reward (especially penalizing omission when task reward is zero) are clever mechanisms.

3. Good empirical performance, especially for a 8B/4B model

4. The authors provide theoretical grounding by providing the proof that the omission policy is upperbounded by the KL



Weaknesses

1. For many practical applications, the massive computational overhead of training (hundreds of GPU hours per task) may significantly outweigh the tokens saved during inference.

2. The theoretical analysis in Section 5 relies on highly abstract definitions of "semantic distance" and Lipschitz constants without clearly linking them to the practical implementation of LLM agents.

3. Removing critical context breaks continuity, requiring extra reasoning tokens to recover information and losing out on efficiency.

4. I agree with the authors themselves that the training pipeline is fragile. Over-tuning the initial supervised phase can cause a total policy collapse during RL and hence they enforce a strict single-epoch limit to prevent performance degradation on complex tasks. It can limit RL on complex tasks. It can over-optimize for token savings (reward hacking).

---

> ### Author Rebuttal · Authors · 2026-03-31
>
> > [W1]
>
> We would like to clarify that the "hundreds of GPU hours" refers to the **total training cost across all agent environments**, not per task (e.g., DeepSearch 32h, WebShop 36h shown in Line 976).
>
> Unlike the "large-size + large-scale training" paradigm (e.g., OpenAI o3), Agent-Omit uses a "small-size model + lightweight training" approach. Training is a modest, one-time fixed cost, while inference savings are recurring. Based on official API pricing:
>
> **Table R1: Cost Comparison (on WebShop)**
>
> |Metric|Agent-Omit-8B|OpenAI o3|
> -|-|-
> Training Cost (one-time)|$416|—
> Inference Tokens (200 queries)|1,561,310|—
> Inference Cost (200 queries)|$0.25|$8.85
> **Training + Inference Cost (10K queries)**|**$439**|**$442.5**
>
> The key insight is: **Agent-Omit-8B achieves superior accuracy to frontier models at a fraction of the per-query cost**. The break-even point against OpenAI o3 is reached at ~10K queries, making this one-time training investment highly practical for real-world applications.
>
> > [W2&Q1]
>
> We would like to clarify that the "semantic distance" in our theoretical analysis is **not an abstract, unmeasurable quantity** — it corresponds to the distance between LLM-generated token sequences in the **embedding space** (i.e., cosine distance between the LLM's output). This is computable in practice shown in Line 568.
>
> More importantly, this semantic distance is explicitly **connected to the KL divergence in our RL training process**. By constraining the KL divergence during GRPO training, we are effectively bounding this semantic distance in the representation space, ensuring the model's omission behavior does not deviate significantly from the reference policy. We will add a paragraph in Section 5 explicitly connecting the semantic distance $d(y, y')$ to the KL divergence term $D_{KL}(\pi_\theta \| \pi_{ref})$ in our GRPO objective (Equation 4), showing how bounding the latter implicitly controls the former.
>
> > [W3]
>
> We fully agree that removing critical context breaks continuity and degrades performance — This aligns exactly with our findings in Section 3.1: removing either the initial or final-turn reasoning context reduces both accuracy and efficiency.
>
> Our dual-stage training prevents this:
> - **Data Collection (SFT stage)**: We explicitly discard samples where omission leads to a loss of efficiency or accuracy. The model is only trained on verified correct trajectories **after removing context**.
> - **Reward Design (RL stage)**: If the agent incorrectly omits a critical turn, it will fail the task ($R_{task} = 0$). Our reward mechanism severely penalizes this by enforcing $R_{omit} = 0$, ensuring such detrimental omissions receive a **negative training signal**.
>
> Thanks to this training design, Agent-Omit-8B-RL achieves substantial efficiency gains without sacrificing performance. As demonstrated in Table 3, it consistently achieves the highest Pass@1 accuracy while incurring the lowest average token cost.
>
> Furthermore, our omission behavior analysis in Figure 6 also proves that this efficiency is driven by selective omission. As can be seen, omissions peak during intermediate turns (Turns 3-10), but the agent heavily avoids omission during critical turns (initial planning turn and final generation turn).
>
> These results demonstrate that **our selective context omission (context removal) improves both agent effectiveness and efficiency**.
>
> > [W4]
>
> The single-epoch RL is not a fragility but a **deliberate design choice**. We conducted additional experiments on WebShop with RL epochs increased to 2, 4, and 6:
>
> **Table R2: RL Epoch Ablation on WebShop**
>
> |Task|RL Epoch|Pass@1 (%)|Avg Token|
> -|-|-|-
> WebShop|1|23.57|8,764
> WebShop|2|23.21|8,698
> WebShop|4|22.85|8,721
> WebShop|6|22.43|8,759
>
> As shown, increasing RL epochs leads to marginal accuracy degradation while token savings remain largely unchanged. This indicates that over-tuning provides diminishing returns, confirming that 1 epoch is empirically optimal.
>
> Regarding reward hacking: our design enforces $R_{omit} = 0$ when $R_{task} = 0$, which means the agent **cannot achieve higher efficiency reward by sacrificing task accuracy**. This hard constraint effectively prevents reward hacking.
>
> > [Q2]
>
> Thank you for this clarification question. The Qwen3-8B Base and our Agent-Omit model use the **same system prompt** shown in Section B.1. The only difference is that Agent-Omit has been trained with our SFT + RL pipeline, while the Base model has not. Our aim here is precisely to **validate whether our proposed pipeline is meaningful** — the performance improvement confirms that the efficiency gains stem directly from our training method, rather than prompt engineering. We will make this explicit in the revised version.

---

> > ### Author Rebuttal · Reviewer_Dzb5 · 2026-04-04
> >
> > Thanks for the thorough rebuttal. The cost break-even analysis helps, though the 10K-query assumption is optimistic for many real-world settings. The prompt clarification for Qwen3-8B Base also clears things up. However, my concerns about the theoretical contribution remain. Changing Lemma 5.1 to an assumption confirms its limited novelty. I will maintain my current score.

---

> > > ### Author Response · Authors · 2026-04-04
> > >
> > > Dear Reviewer Dzb5,
> > >
> > > Thank you for acknowledging our rebuttal and for the constructive feedback throughout this process. We are glad that the cost break-even analysis and the Qwen3-8B Base prompt clarification were helpful.
> > >
> > > We would like to respectfully address your remaining concern regarding theoretical contribution.
> > >
> > > **Clarification on the theoretical contribution**: We agree that renaming Lemma 5.1 to Assumption A.1 better reflects its nature. However, we want to clarify that **the theoretical contribution of our paper is not this lemma. Our main theoretical result is Theorem 5.2**, which proves that the performance deviation of the omission policy is upper-bounded by the KL divergence between the learned policy and the reference policy. This result provides **a formal guarantee that the omission behavior will not cause catastrophic policy drift** — as long as the KL divergence is controlled during training.
> > >
> > > **Practical value of Theorem 5.2**: While we agree this does not guarantee convergence to the optimal policy, it provides a practically actionable insight: the KL coefficient β in our GRPO objective (Eq. 4) directly serves as a trust region to control the omission–performance trade-off. By bounding the KL divergence during training, we effectively bound the deviation of the omission policy from the reference policy, **providing a principled safeguard for the agent's adaptive omission capability**. This is not merely a mathematical artifact — it directly informs how we train the agent in practice.
> > >
> > > **Contribution scope**: We respectfully note that **our paper is not a pure theoretical paper** — the theoretical analysis occupies less than half a page in the main text (Line 252-Line 289, Section 5) and is intended to provide analytical support rather than serve as a standalone contribution. **Our primary contributions lie in** (1) the preliminary study (Section 3.1), which provides both qualitative and quantitative analyses of how thought and observation affect agent effectiveness and efficiency across turns, and (2) the training framework (cold-start data synthesis + omit-aware RL). The theory serves as a principled foundation for the training choices — analogous to how PPO's clipping mechanism is theoretically motivated by the trust region framework but valued primarily for its practical impact. We believe this empirical-insight-plus-methodology-plus-theory paradigm is well-suited for the ICML community.
> > >
> > > We hope you may reconsider whether the comprehensive empirical validation (5 benchmarks, 2 model sizes) together with the theory-grounded framework design meets the bar for ICML acceptance.
> > >
> > > Thank you again for your thorough and constructive review.

---

### Decision · Program_Chairs · 2026-04-30

**Decision:**

Accept (regular)

**Comment:**

This paper provides an effective training framework for adaptive context omission. There is a consensus among the reviewers and me that the idea is smart and well-motivated. There are deliberations on the scalability, the cost, and the theory of the work. The reviewers and I eventually buy the story that 1) the empirical results that successfully scale to 120B models and 2) the theory part is not that much important so it might not jeopardize the work. I therefore recommend accept should the ICML program have enough space, despite the marginal average score.

However, I notice that the anonymous link does not contain any publicly available code. I tend to trust the authors on this, but the authors MUST make the code fully open and fully executable by the camera-ready deadline, as the condition that I maintain my recommendation.